# DivIVA Phosphorylation Affects Its Dynamics and Cell Cycle in Radioresistant *Deinococcus radiodurans*

Reema Chaudhary,[a,b] Swathi Kota,[a,b] Hari S. Misra[a,b,c]

[a]Molecular Biology Division, Bhabha Atomic Research Centre, Mumbai, India
[b]Life Sciences, Homi Bhabha National Institute, Mumbai, India
[c]School of Science, GITAM, Visakhapatnam, Andhra Pradesh, India

**ABSTRACT** DivIVA is a member of the Min family of proteins that spatially regulates septum formation at the midcell position and cell pole determination in *Bacillus subtilis*. *Deinococcus radiodurans*, a Gram-positive coccus-shaped bacterium, is characterized by its extreme resistance to DNA-damaging agents, including radiation. *D. radiodurans* cells exposed to gamma radiation undergo cell division arrest by as-yet-uncharacterized mechanisms. *divIVA* is shown to be an essential cell division gene in this bacterium, and DivIVA of *D. radiodurans* (drDivIVA) interacts with genome segregation proteins through its N-terminal region. Earlier, RqkA, a gamma radiation-responsive Ser/Thr quinoprotein kinase, was characterized for its role in radioresistance in *D. radiodurans*. Here, we showed that RqkA phosphorylates drDivIVA at the threonine 19 (T19) residue. The phospho-mimetic mutant with a mutation of T19 to E19 in DivIVA (DivIVAT19E) is found to be functionally different from the phospho-ablative mutant (DivIVAT19A) or the wild-type drDivIVA. A DivIVAT19E-red fluorescent protein (RFP) fusion expressed in the wild-type background showed the arrest in the typical dynamics of drDivIVA and the loss of its interaction with the genome segregation protein ParA2. The allelic replacement of *divIVA* with *divIVA^T19E^-rfp* was not tolerated unless drDivIVA was expressed episomally, while there was no phenotypic change when the wild-type allele was replaced with either *divIVA^T19A^-rfp* or *divIVA-rfp*. These results suggested that the phosphorylation of T19 in drDivIVA by RqkA affected its *in vivo* functions, which may contribute to the cell cycle arrest in this bacterium.

**IMPORTANCE** *Deinococcus radiodurans*, a radioresistant bacterium, lacks LexA/RecA-mediated DNA damage response and cell cycle regulation as known in other bacteria. However, it adjusts its transcriptome and proteome upon DNA damage. In eukaryotes, the DNA damage response and cell cycle are regulated by Ser/Thr protein kinases. In *D. radiodurans*, we characterized a gamma radiation-responsive Ser/Thr quinoprotein kinase (RqkA) that phosphorylated DNA repair and cell division proteins in this bacterium. In previous work, the effect of S/T phosphorylation by RqkA on activity improvement of the DNA repair proteins has been demonstrated. This study reports that Ser phosphorylation by RqkA attenuates the function of a cell polarity and plane of cell division-determining protein, DivIVA, and its cellular dynamics in response to DNA damage, which might help to understand the mechanism of cell cycle regulation in this bacterium.

**KEYWORDS** cell cycle, cell polarity, coccus bacterium, DivIVA, DNA damage response, radiation resistance, S/T phosphorylation, time-lapse microscopy, radioresistance

Proteins undergo several types of posttranslational modifications (PTMs), albeit the majority of them have been reported in eukaryotes. Among these, the most common PTMs are phosphorylation, acetylation, methylation, carbonylation, glycosylation, and ubiquitination (1). The active presence of cognate kinases and phosphatases in

Address correspondence to Hari S. Misra, hsmisra@barc.gov.in.

The authors declare no conflict of interest.

certain ratios determines the phosphoprotein homeostasis in the cells. Different types of protein phosphorylation have been reported in different organisms, and the selectivity of their functions has been shown in different organisms. For example, Ser/Thr/Tyr phosphorylation is predominantly studied in the DNA damage response and cell cycle regulation in eukaryotes. Histidine kinases as a part of two-component systems are best characterized as stress response kinases in bacteria (2–4). However, understanding the roles of Ser/Thr (S/T) protein kinases (STPKs) in the maintenance of genome integrity and regulation of cell division in bacteria has also gained a considerable importance. For instance, a eukaryotic-type STPK was first identified in the Gram-negative bacterium *Myxococcus xanthus* (5). STPKs have now been identified in both biotic- and abiotic-stress-tolerant bacteria, including pathogens like *Pseudomonas aeruginosa* (6), *Enterococcus faecalis* (7), *Staphylococcus aureus* (8), *Mycobacterium tuberculosis* (9), *Streptococcus pneumoniae* (10), *Streptococcus agalactiae* (11), *Streptococcus pyogenes* (12), and the superbug-like *Deinococcus radiodurans* (13, 14). S/T phosphorylation has been reported in bacterial proteins involved in several biological processes, including genome functions, virulence, and stress responses (14–17). The S/T phosphorylation has also been detected in bacterial cell division proteins, and the effects of such phosphorylation on the functions of FtsZ have been characterized in many bacteria (18, 19). DivIVA undergoes phosphorylation by StkP in *S. pneumoniae* and *Streptococcus suis*, and the effects of such phosphorylation on the regulatory roles of this protein in cell division and morphogenesis have been reported (20–22). In *Mycobacterium*, Wag31 (a homolog of DivIVA) is involved in septum formation, cell wall synthesis, and/or chromosome segregation and is found to be a substrate for the protein kinases PknA and PknB (23).

*D. radiodurans* is characterized by its extraordinary resistance to DNA-damaging agents, including radiation and desiccation (24–26). Although the growth of this bacterium is arrested upon exposure to gamma radiation, the levels of the majority of cell division proteins do not change during postirradiation recovery (PIR) (27). Interestingly, the RecA/LexA-mediated canonical SOS response, which is synonymous with the DNA damage response and cell cycle regulation in the majority of bacteria, is lacking in this bacterium (28–30). However, *D. radiodurans* modulates its proteome and transcriptome in response to DNA damage (31, 32). A few mechanisms that partly complement the absence of the LexA/RecA-mediated DNA damage response and cell cycle regulation have been suggested. For instance, DdrO/PprI (IrrE) regulate genome function by mimicking LexA/RecA protein functions (33–35). It is also known that guanine quadruplex (G4) DNA structure dynamics regulate genome function in response to gamma radiation-induced damage (36–38). These examples could explain the DNA damage-responsive regulation of the expression of those DNA repair proteins that have the respective regulatory signatures. However, these mechanisms cannot explain the functional regulation of cell division proteins during postirradiation recovery in this bacterium. Independently, a DNA damage-responsive Ser/Thr quinoprotein kinase (RqkA) has been characterized for its role in radiation resistance and double-strand break (DSB) repair in *D. radiodurans* (39). RqkA phosphorylates a number of substrates of this bacterium, including PprA, RecA, Hu, TopoIB, RecD, and UvrD (26). The effect of S/T phosphorylation on the activity regulation of RecA, PprA, and FtsZ has been demonstrated (40–42). DivIVA of *D. radiodurans* (drDivIVA) has been functionally characterized for its interaction with cell division and genome maintenance proteins, and *divIVA* was found to be an essential gene in this bacterium (43–45). Here, for the first time, we report the phosphorylation of drDivIVA by RqkA and further demonstrate its significance in the regulation of drDivIVA functions. The phosphorylation of drDivIVA by RqkA and the essential nature of drDivIVA raised the question of whether DivIVA phosphorylation conditions the cell to undergo growth arrest in response to DNA damage. We showed that drDivIVA was phosphorylated by the DNA damage-responsive STPK RqkA, and furthermore, that the subcellular role of drDivIVA in spatial regulation was arrested when phosphorylation at residue T19 was mimicked with glutamate,

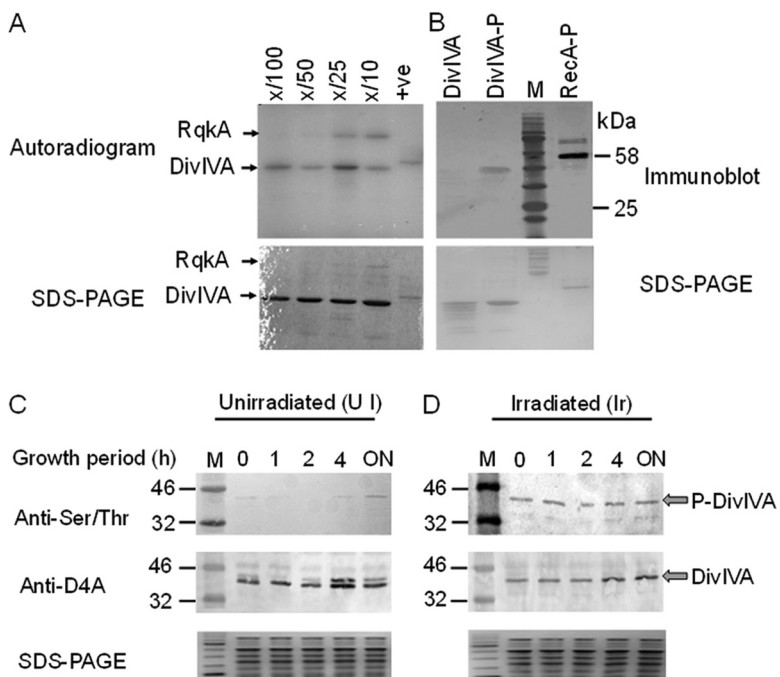

**FIG 1** Phosphorylation of drDivIVA by RqkA. (A) drDivIVA was incubated with increasing concentrations of RqkA (X/100, X/50, X/25, and X/10, where X is DivIVA) in kinase buffer containing [γ-³²P]ATP, the reaction mixtures were separated on 12% SDS-PAGE gels, and an autoradiogram was developed. (B) *E. coli* BL21 cells expressing RqkA (pRadRqkA) and drDivIVA (pETD4A) separately and together (pRadRqkA + pETD4A) in the BL21 host. The drDivIVA proteins routed through cells expressing RqkA (DivIVA-P) and vector-harboring cells (DivIVA) were run on SDS-PAGE gels and checked for phosphorylation signals using anti-phospho-serine/threonine antibodies. RecA routed through RqkA-expressing cells was used as a positive (+ve) control (RecA-P). (C, D) *In vivo* phosphorylation of drDivIVA was checked in *D. radiodurans* cells expressing histidine-tagged DivIVA on pRGhisD4A. The deinococcal cells were grown before being exposed to 6 kGy gamma radiation. Both the treated cells and the sham control (unirradiated, but kept on ice) were allowed to grow, and aliquots were collected at different postirradiation recovery (PIR) times (0, 1, 2, and 4 h and overnight [ON]). The recombinant protein was partially purified from the sham control (C) and irradiated (D) cell lysates (SDS-PAGE) using an affinity tag and running on 12% SDS-PAGE. Proteins were blotted on membranes and hybridized using anti-DivIVA and anti-phospho-Ser/Thr antibodies (Anti-Ser/Thr). M, molecular size marker (kDa). The data shown are representative of a reproducible experiment repeated three times.

suggesting the possible involvement of drDivIVA phosphorylation in cell cycle arrest when *D. radiodurans* is exposed to gamma radiation.

(This work was a part of the doctoral thesis of Reema Chaudhary, submitted to the Homi Bhabha National Institute [Department of Atomic Energy—deemed to be a university], Mumbai, India).

## RESULTS

**RqkA phosphorylates drDivIVA at the T19 residue.** The conserved motif "S/T-Q-X-hydrophobic-hydrophobic" (where X is any amino acid except the positively charged residues) in drDivIVA is predicted to be the site of phosphorylation by a eukaryotic-like Ser/Thr protein kinase (eSTPK), such as RqkA. The recombinant drDivIVA, when incubated with RqkA in the presence of [γ-³²P]ATP, produced 2 phospho-bands, corresponding to drDivIVA and the autophosphorylated form of RqkA, respectively (Fig. 1A). drDivIVA phosphorylation by RqkA was further confirmed *ex vivo* in transgenic *E. coli* BL21 cells coexpressing both RqkA and DivIVA (see Fig. S1G at https://barc.gov.in/publications/mbio/DivIVA/figures_s.pdf). The recombinant drDivIVA was purified from two backgrounds of *E. coli* BL21 cells, of which one expressed only DivIVA (pETD4A) and the other coexpressed RqkA (pRadRqkA) and DivIVA (pETD4A). The phosphorylation status of the protein was verified using polyclonal anti-phospho-Ser/Thr epitope antibodies (Cell Signaling, Inc.). The

recombinant phosphorylated RecA (RecA-P) purified from the cells coexpressing RqkA was used as a control. The DivIVA-P and RecA-P coexpressed in the RqkA background showed phosphorylation, while drDivIVA obtained from *E. coli* cells harboring only pETD4A did not show S/T phosphorylation (Fig. 1B).

The phosphorylation of drDivIVA was monitored in *D. radiodurans* cells grown under unirradiated and gamma-irradiated conditions. For that, the cells expressing histidine-tagged drDivIVA episomally were lysed, the separated proteins were immunoblotted with antibodies against Ser/Thr epitopes, and the identity of DivIVA was ascertained using drDivIVA antibody. The results showed growth phase-dependent phosphorylation under normal conditions (Fig. 1C). For instance, drDivIVA was not phosphorylated in the early phase of growth but showed detectable phosphorylation in stationary-phase cells under normal conditions. Phosphorylation was detected in response to gamma radiation in both early postirradiation recovery (PIR) periods that witnessed growth arrest and in stationary-phase cells recovered overnight (Fig. 1D). These results confirmed that drDivIVA is phosphorylated by RqkA *in vitro* and is a Ser/Thr phosphoprotein in *D. radiodurans*. The phosphorylated drDivIVA was purified from bacterial cells, while the non-phospho version was isolated from cells containing only the expression vector (pETD4A). Both proteins were subjected to mass spectrometric analysis (see Fig. S1A to C at https://barc.gov.in/publications/mbio/DivIVA/figures_s.pdf). As expected, no phospho-residue(s) were found in the drDivIVA nonphospho preparation. The mass-spectrometric analysis detected a single phosphorylation site at threonine 19 (T19) in drDivIVA. Previously, RecA and PprA showing very high levels of phosphorylation by RqkA were detected with multiple sites of phosphorylation (40, 41).

In *B. subtilis*, amino acids arginine-18 and glycine-19 play a role in the targeting of DivIVA to the cell poles, and hence, these residues constitute a polarity-determining motif (46–48). To better understand the significance of drDivIVA T19 phosphorylation, the phosphorylation motif was examined using pairwise alignment with *B. subtilis* DivIVA. Interestingly, the N-terminal sequence of drDivIVA was found to be largely conserved with its numerical value of 9, and the T19 residue was found to be the neighbor of the polarity-determining motif of this protein (see Fig. S1C at https://barc.gov.in/publications/mbio/DivIVA/figures_s.pdf). Thus, the possibility of T19 playing a role in pole determination in cocci and the effect of phosphorylation on its function were hypothesized and investigated. Subsequently, the T19 residue in drDivIVA was replaced with alanine or glutamic acid to yield DivIVA$^{T19A}$ (T19A below) and DivIVA$^{T19E}$ (T19E below), and these phospho-mutants were analyzed for phosphorylation by RqkA *ex vivo*. The results showed that both the T19A and T19E mutants were phospho negative when incubated with RqkA (see Fig. S1E at https://barc.gov.in/publications/mbio/DivIVA/figures_s.pdf). These results provided strong evidence that drDivIVA is a Ser/Thr-type phosphoprotein in *D. radiodurans* and is phosphorylated at the T19 residue.

**The T19E mutation in drDivIVA alters the wild-type pattern of localization in *D. radiodurans*.** To understand the effect of T19 phosphorylation on the cellular dynamics of drDivIVA, the *in vivo* localizations of both the wild-type (WT) and phospho-mutant forms of drDivIVA with respect to FtsZ were monitored. C-terminally tagged red fluorescent protein (RFP) fusions of drDivIVA (DivIVA-RFP) and its mutants and a green fluorescent protein (GFP)-tagged fusion of FtsZ (FtsZ-GFP) were coexpressed in *D. radiodurans* R1. The localization patterns of both proteins were examined by confocal microscopy (Fig. 2A and B). The patterns of FtsZ localization were very similar in all three cell types expressing drDivIVA or its phospho-mutants episomally in a wild-type background. For instance, the FtsZ ring was seen in a closed conformation in nearly 48.27% ± 5.41% (mean ± standard deviation) to 50.1% ± 4.10% of the cell populations, while 43.34% ± 4.89% to 57.17% ± 5.72% of the cell populations had the FtsZ ring in an open conformation in all three cell types (DivIVA$^{WT}$, DivIVA$^{T19A}$, and DivIVA$^{T19E}$). It may be mentioned that all three cell types had the wild-type copy of drDivIVA in the background, and the possibility that that contributed to the nearly complete lack of change in the localization pattern of FtsZ cannot be ruled out.

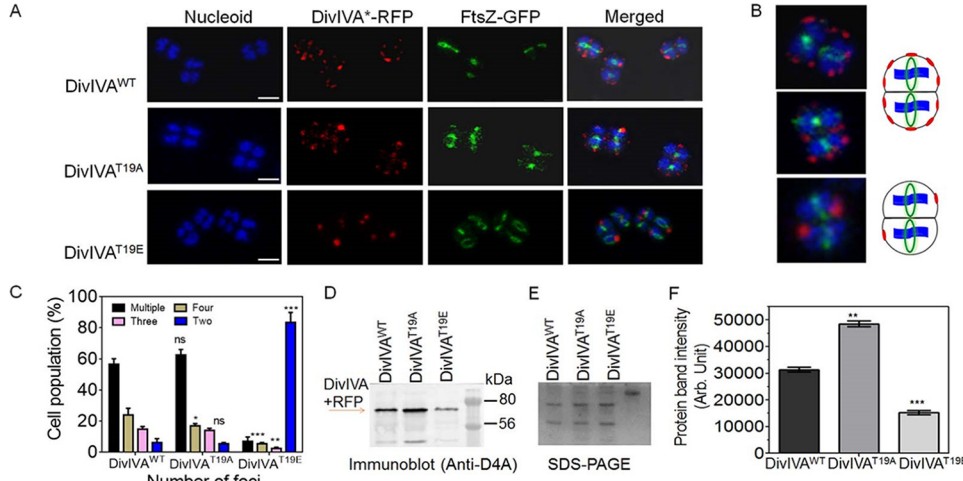

**FIG 2** Localization of wild-type and phospho-mutant forms of drDivIVA in *D. radiodurans*. (A) Fluorescence imaging of *D. radiodurans* cells expressing DivIVA$^{WT}$, DivIVA$^{T19A}$, and DivIVA$^{T19E}$ under the constitutive promoter on the pRADgro plasmid. The cells were observed for nucleoid staining (blue), FtsZ-GFP localization (green), and DivIVA*-RFP, where the asterisk indicates drDivIVA variants (DivIVA$^{WT}$, DivIVA$^{T19A}$, and DivIVA$^{T19E}$); the merged panel shows the image combining all three channels. Scale bars = 2 $\mu$m. (B) Zoomed-in versions of representative cells are shown for clarity. (C) The cells were counted for the numbers of foci. (D) The total proteins of *D. radiodurans* R1 cells harboring pRGD4A (DivIVA$^{WT}$), pRGD4A$^{T19A}$ (DivIVA$^{T19A}$), and pRGD4A$^{T19E}$ (DivIVA$^{T19E}$) were separated on SDS-PAGE (E) and immunoblotted with anti-DivIVA antibodies. (F) The intensities of protein bands were measured using Image J and plotted using GraphPad Prism software. Error bars show standard deviations ($n$ = 12). Statistical analyses were done using Student's *t* test. **, $P \leq 0.01$; ***, $P \leq 0.005$. Data shown without statistical analysis are representative of reproducible results repeated at least three times.

As far as the localization patterns of drDivIVA and its phospho-mutants are concerned, multiple DivIVA$^{WT}$ foci were seen at the cell periphery, and FtsZ foci were observed in the space(s) unoccupied by drDivIVA, suggesting its role in the spatial positioning of the FtsZ ring in *D. radiodurans*. A careful examination revealed that the cells that had either completed cell division or had just begun it contained fewer than four foci of drDivIVA per cell. Nearly 56.5% $\pm$ 3.536% of the cells with multiple foci had them placed in juxtaposition in the two compartments of the tetrad, indicating that the dynamics of this cell division protein was synchronized. Furthermore, nearly 62.5% $\pm$ 3.965% of cells expressing DivIVA$^{T19A}$-RFP protein showed more than four foci and 14% $\pm$ 1.414% had less than four foci, while the majority (83.6% $\pm$ 6.364%) of cells expressing DivIVA$^{T19E}$-RFP protein showed two foci per cell (Fig. 2C). Since the cumulative intensity of T19E foci appeared different than the cumulative intensities of T19A and drDivIVA foci separately, the possibility of this phenotype being due to the change in protein levels of DivIVA phospho-mutants was argued. The levels of DivIVA protein in the cells expressing drDivIVA and mutants (T19A and T19E) were measured by immunoblotting using protein-specific antibodies (Fig. 2D and E). Interestingly, the levels of episomally expressed drDivIVA and its mutant forms were found to be different. For example, the level of the T19E protein was discovered to be lower than those of the wild-type and T19A proteins (Fig. 2E). The difference in the levels of drDivIVA variants, despite being under identical promoters, translation machinery, and translation fusion, is intriguing, and a possibility of phosphorylation-mediated homeostasis of drDivIVA in *D. radiodurans* could be speculated, which would be worth studying independently. Nevertheless, these results suggested that RqkA phosphorylation at the T19 residue in drDivIVA affects the cellular localization of this protein in *D. radiodurans*.

**Mimicking T19 phosphorylation in drDivIVA affects its spatial placement with respect to FtsZ in *D. radiodurans*.** The *D. radiodurans* cells expressing drDivIVA-RFP and FtsZ-GFP on plasmids were subjected to Z-stacking for 15 to 20 slices of 200 nm in each plane, and the localizations of both proteins with respect to each other were monitored. Several single cells were analyzed at different angles to understand the planes of localization of both the proteins. The results shown here are for a representative

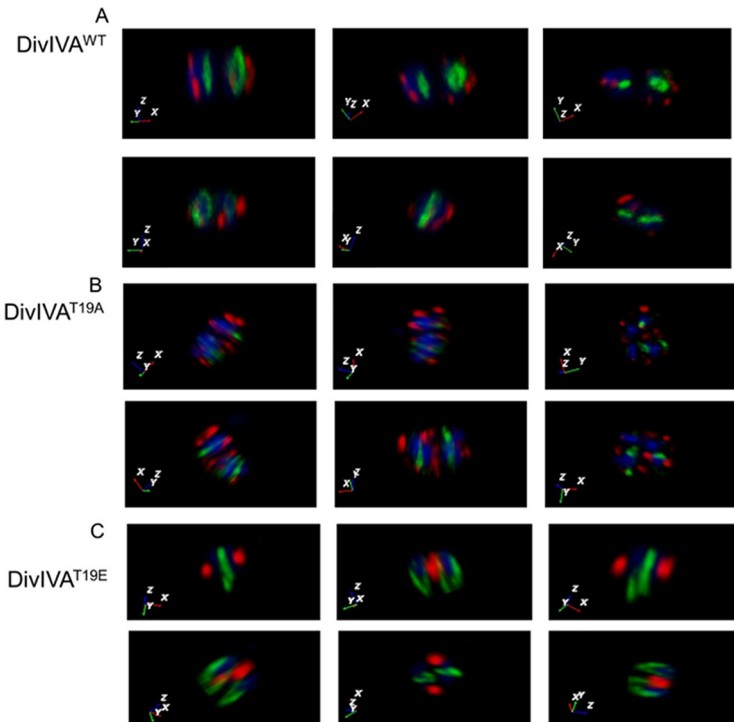

**FIG 3** The localization of drDivIVA/phospho-mutants and FtsZ in *D. radiodurans*. *D. radiodurans* cells expressing wild-type drDivIVA-RFP (DivIVA^WT), its phospho-mutants DivIVA^T19A and DivIVA^T19E (red), and FtsZ-GFP (green) were analyzed for the planar positions of these proteins and the genome (blue) using confocal microscopy. The images were taken at several Z-planes and superimposed to create the overall 3-D picture, and a single cell was analyzed by rotating at various angles. The *x-y-z* axes are shown in the bottom-left corners, denoting the positions of drDivIVA, FtsZ, and the genome, respectively. The *x-y-z* axes are color coded according to the colors of DivIVA (red), FtsZ (green), and the nucleoid (blue). The data shown are representative of a reproducible experiment repeated at least three times.

cell from the pool of identical cells (Fig. 3). They showed that both proteins were localized in different planes with respect to each other and the nucleoid. For instance, drDivIVA foci appeared both in conjunction with and away from the nucleoid, whereas FtsZ foci appeared to be perpendicular to the nucleoid, as shown by the *x-y-z* axis in the corresponding image (Fig. 3A). The perpendicularity of the Z-ring with respect to the genome is well known in rod-shaped bacteria. The drDivIVA protein localizes in a plane different than that of FtsZ, and this differential localization of both proteins highlights the formation of an FtsZ ring in the plane that is free from drDivIVA, thereby determining the plane of division in the coccus-shaped bacteria. In *B. subtilis*, DivIVA localizes at new division sites that would form new poles in the daughter cells (49). In terms of pattern of localization, the T19A mutant appeared to be similar to the wild type (Fig. 3B). However, the T19E mutant showed localization in a different pattern, and the arrangement was not throughout the membrane but, rather, at some predefined position, and it lost its dynamic nature in the cells during growth (Fig. 3C). These results suggested that RqkA phosphorylation of drDivIVA affected the wild-type function of this protein in the cell division of *D. radiodurans*.

**Allelic replacement shows intolerance of the phospho-mimetic allele under normal growth conditions.** The *divIVA* gene is shown to be essential for the normal growth of this bacterium, and its chromosomal copy could be replaced with a selection marker only when drDivIVA was expressed episomally (44). The effect of phosphorylation on drDivIVA functions was further evaluated by an allelic replacement of the wild-type copy with the phospho-mutant alleles in the chromosome of this bacterium. The chromosomal copy of *divIVA* could be replaced with *divIVA-rfp* in *D. radiodurans*, and the expression of drDivIVA-RFP under its native promoter was ascertained by fluorescence microscopy (Fig. 4A). These cells showed normal growth and produced multiple

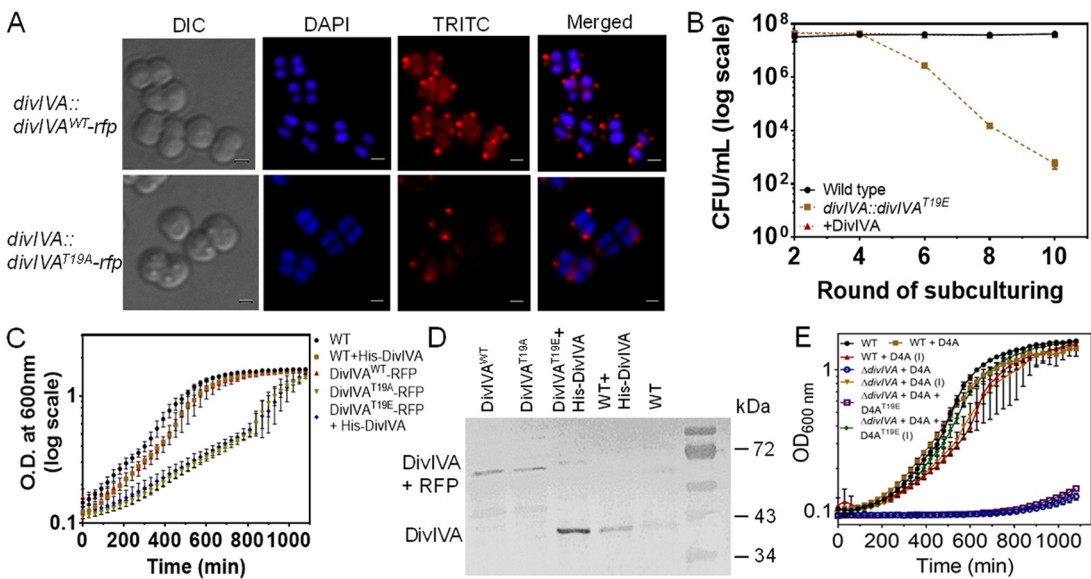

**FIG 4** Chromosomal integration of *divIVA-rfp* and its variants under the native promoter of *divIVA* in *D. radiodurans*. The drDivIVA coding sequences in the chromosome of *D. radiodurans* were replaced with RFP fusions of wild-type and phospho-mutants of drDivIVA, as described in Materials and Methods. (A) The successful replacement of the wild-type protein with DivIVA-RFP and DivIVA^T19A^-RFP was confirmed by fluorescence microscopy. The pNKD4A^T19E^ plasmid was linearized and transformed into *D. radiodurans*, and transformants were scored in the presence of kanamycin. These cells were subcultured for several rounds in the presence of kanamycin. Scale bars = 1 μm. (B) The possible replacement of *divIVA* with *divIVA^T19E^-rfp-nptII* was monitored on an agar plate supplemented with antibiotic after the 1st, 3rd, 5th, 7th, 9th, and 11th rounds of subculture and compared with the Δ*parA* mutant as a positive control. During these rounds of subculture, cells with wild-type *divIVA* and its prospective phospho-mimetic knock-in mutant were plated on TGY and TGY + kanamycin (8 μg/mL) agar plates, respectively. The CFU were obtained and plotted as a function of a round of subculture. (C) Growth-curve kinetics of wild-type (WT) cells, cells with knock-in derivatives *divIVA-rfp*, (DivIVA^WT^-RFP), and *divIVA^T19A^-rfp* (DivIVA^T19A^-RFP), and cells with *divIVA^T19E^* expressing His-DivIVA episomally (DivIVA^T19E^-RFP + His-DivIVA), as well as wild-type cells harboring pRGhisD4A (WT + His-DivIVA), were monitored. (D) The total proteins of these knock-in derivatives of *D. radiodurans* were separated on SDS-PAGE gels and immunoblotted with anti-DivIVA antibodies. (E) The dependence of wild-type-copy replacement with the *divIVA^T19E^* allele on episomal expression of drDivIVA was further ascertained by conditional expression of the wild-type protein. Error bars show standard deviations. Data shown without statistical attributes are representative of reproducible experiments repeated at least three times.

drDivIVA-RFP foci that were similar to those in the cells expressing drDivIVA-RFP episomally, albeit with a different intensity. For studying the cellular dynamics of T19A and T19E proteins expressed under the native promoter, replacement of the wild-type copy with *divIVA^T19A^-rfp* and *divIVA-rfp* alleles was attempted. The replacement of *divIVA* with *divIVA^T19A^-rfp* could be achieved, and these cells showed normal growth and multiple foci of DivIVA^T19A^-RFP per cell (Fig. 4A). Notably, homogeneous replacement of *divIVA* with *divIVA^T19E^-rfp* could not be achieved, and the cells began to be debilitated as they approached homogeneity and eventually stopped growing under normal conditions. The CFU monitored during successive rounds of subculture showed a gradual decrease in prospective *divIVA^T19E^-rfp* cells (Fig. 4B). However, integration of the *divIVA^T19E^-rfp* allele was achieved when drDivIVA was expressed in *trans* on a plasmid. Interestingly, these cells grew more slowly than cells expressing T19A and wild-type controls (Fig. 4C). Surprisingly, the cells integrated with the *divIVA^T19E^-rfp* allele in the presence of an episomal drDivIVA also showed a relatively lower amount of DivIVA^T19E^-RFP protein, further indicating that phosphorylation seems to be deleterious for the stability of this protein in *D. radiodurans* (Fig. 4D). The results presented here suggest that phosphorylation of drDivIVA at the T19 position seems to make this protein nonfunctional or less stable, and therefore, the *D. radiodurans* cells harboring *divIVA^T19E^-rfp* in the place of *divIVA* could not be maintained under normal conditions.

**drDivIVA marks the successive planes of cell division and regulates the position of FtsZ ring formation.** The cellular dynamics of drDivIVA (DivIVA-RFP) and FtsZ (FtsZ-GFP) were monitored in *D. radiodurans* cells growing under normal conditions. Microscopic examination showed a cyclic change in the typical shapes of the FtsZ ring,

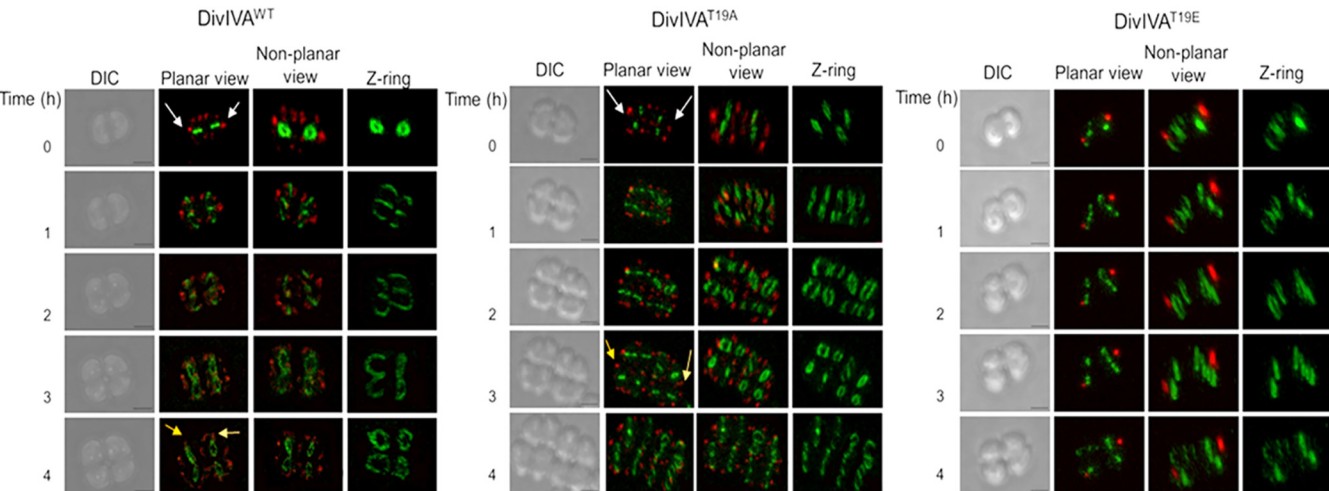

**FIG 5** Time-lapse microscopy of the *D. radiodurans* cells expressing drDivIVA-RFP derivatives and FtsZ-GFP. *D. radiodurans* cells expressing the RFP fusion of wild-type drDivIVA (DivIVA^WT), DivIVA^T19A, and DivIVA^T19E along with FtsZ-GFP were allowed to divide on microscopic slides and observed at different time points. Images were taken in differential inference contrast (DIC), GFP, and RFP channels and are presented in planar view, as well as in 3-D views as nonplanar views. The scale bars for all of the imaging were kept constant at 500 nm. Cells that are proceeding toward the final stage of division and have had drDivIVA observed in the mother cell appear in the daughter cells are marked with white arrows. A representative set of data from a reproducible experiment is shown.

which correspond to different stages of the cell cycle (see Fig. S3 at https://barc.gov.in/publications/mbio/DivIVA/figures_s.pdf). For instance, the cells preparing for division had the closed form of a Z-ring (Fig. 5, left, $t = 0$ h), which opened to a 3-shaped form and finally returned to the closed form ($t = 3$ h) in cells producing two daughter cells ($t = 4$ h). Notably, the plane of opening of the FtsZ ring from closed to 3-shaped was found to be perpendicular to that of the closed ring at $t = 0$ h. To our surprise, drDivIVA was localized in a mirror image fashion in the two compartments of the tetrad. As the cell divided, it marked a certain spatial territory, and Z-ring formation occurred in the drDivIVA-free zone ($t = 1, 2$ h). When the cell division progressed toward the final stage, the foci of drDivIVA appeared to spread out in their zone, and the foci that were observed in the mother cell would have passed on to the daughter cells and may possibly have served as memory for the plane of previous division in the mother cells (Fig. 5, white arrows). The daughter cells also possessed drDivIVA foci at the places where recent division had occurred (Fig. 5, yellow arrows; $t = 4$ h). These results showed that the FtsZ ring was produced in drDivIVA free space, and the drDivIVA foci were found at both old and new division sites, which thus pushed the site of FtsZ polymerization to a juxtaposed position in the cells of this bacterium.

**T19E shows distinctive subcellular localization and dynamics compared to those of the wild-type and T19A proteins in *D. radiodurans*.** The effects of drDivIVA phosphorylation on its localization and cellular dynamics, FtsZ nucleation, and the dynamics of FtsZ ring polymorphism were also monitored in the cells expressing T19A and T19E mutants. The T19A mutant was localized at both locations, the new location corresponding to the new plane of cell division and the old location corresponding to the previous division plane. The pattern of T19A localization was found to be very similar to those of wild-type proteins (Fig. 5, DivIVA^WT and DivIVA^T19A). However, the T19E mutants showed contrasting phenotypes compared with those of T19A and wild-type drDivIVA with respect to the FtsZ-GFP conformation in *D. radiodurans* (Fig. 5, DivIVA^T19E). For instance, the opening of the Z-ring at the onset of division occurs in both of the planes, irrespective of the position of T19E and while the cell division is completed, but the foci of the T19E protein did not change their locations. Furthermore, these cells produced only one red-fluorescent spot per cell of the dyad, compared to multiple foci seen in the case of drDivIVA. These cells also showed the cell cycle-dependent FtsZ ring polymorphisms, as observed with wild-type and T19A cells, which could be attributed to the wild-type background with respect to drDivIVA. It may be noted that this study was carried out in a

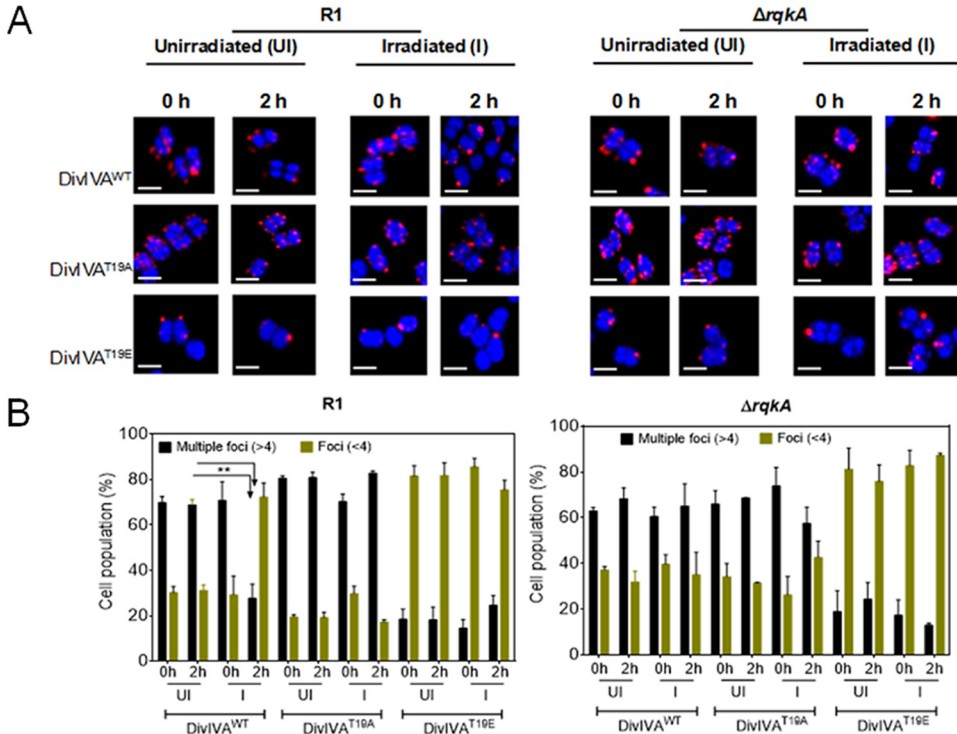

**FIG 6** Localization patterns of DivIVA and its phospho-mutants in both wild-type (R1) and Δ*rqkA* cells grown under unirradiated and irradiated conditions. (A) Wild-type (R1) and Δ*rqkA* cells expressing wild-type (DivIVA^WT), phospho-ablative (DivIVA^T19A), and phospho-mimetic (DivIVA^T19E) derivatives of DivIVA on plasmids pRGD4A, pRGD4A^T19A, and pRGD4A^T19E, respectively, were grown and exposed to 6 kGy gamma radiation (irradiated). A set of these cells were kept on ice as sham controls for irradiation (unirradiated). These cells were allowed to recover (postirradiation recovery [PIR]) as described in Materials and Methods, and cells were collected at 0 h PIR and 2 h PIR and processed for confocal microscopy. Scale bar = 1 μm. (B) Nearly 250 cells from each microscopic field were taken, and the numbers of RFP foci per cell in these populations were counted. A quantitative data set is presented, with statistical analysis. Error bars show standard deviations. **, $P < 0.01$.

wild-type background because of a technical inability (essentiality) to get a chromosomal mutant of *divIVA*. The results obtained here clearly assisted us in understanding the effect of phosphorylation on the dynamics of drDivIVA during normal bacterial growth, clearly indicating that the T19E mutant failed to take part in the process of cell division and, thus, highlighting the importance of drDivIVA phosphorylation in the regulation of cell division in this bacterium.

The significance of gamma radiation-inducible RqkA-mediated phosphorylation in drDivIVA dynamics was monitored in an *rqkA* mutant exposed to gamma radiation. The pattern of drDivIVA localization changed from multiple foci to fewer than four foci after 2 h of irradiation in a wild-type background, whereas it remained unchanged in a Δ*rqkA* background (Fig. 6A). Neither the T19A nor the T19E mutant showed any apparent change in its localization pattern after irradiation in both backgrounds. Interestingly, the localization pattern of drDivIVA upon exposure to gamma radiation changed to nearly that of T19E, as indicated by the smaller number of foci than the multiple foci observed in the cases of T19A and drDivIVA under normal conditions in the wild-type background (Fig. 6B). However, there was no change in the localization patterns of T19A and T19E proteins in both *rqkA* mutant and wild-type backgrounds upon gamma radiation exposure, as well as the drDivIVA pattern in the RqkA-minus background. These results suggested that the gamma radiation responsiveness of RqkA affected drDivIVA functions in this bacterium.

**The localization of drDivIVA seems to be in sync with the segregation of the duplicated genome.** Genome segregation is required for true inheritance of genetic materials into daughter cells through binary fission in bacteria, and the interaction of drDivIVA with its cognate genome segregation proteins has been reported (43). Therefore, the effect of drDivIVA phosphorylation on genome segregation in *D. radiodurans* was monitored by

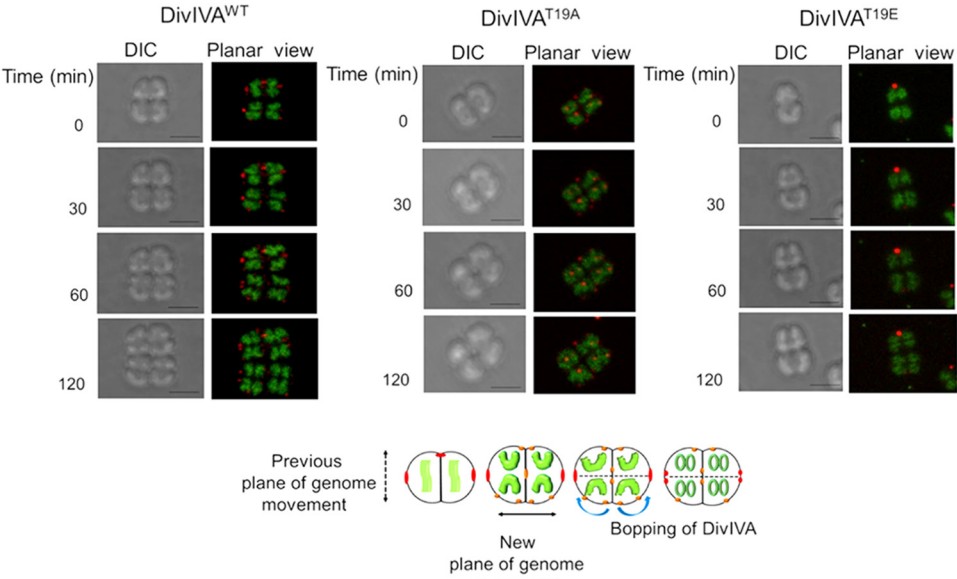

**FIG 7** Top, time-lapse confocal microscopy of *D. radiodurans* cells expressing drDivIVA-RFP and the phospho-mutant forms with respect to genome movement during cell division. *D. radiodurans* cells expressing the RFP fusions of wild-type drDivIVA, DivIVA<sup>T19A</sup>, and DivIVA<sup>T19E</sup> were stained for genomes with SYTO-9 dye and allowed to divide on microscope slides. These cells were imaged in DIC and RFP channels at different time points, and images are presented in DIC and planar views. The scale bars for all the imaging were kept constant at 500 nm. For clarity, representative cells with duplicated copies of the genomes that changed during subsequent stages of cell division are shown. Bottom, a general view of drDivIVA and its mutant dynamics with respect to genome segregation as observed in the majority of cell populations is schematically represented. The data shown are representative of reproducible experiments repeated three times.

time-lapse microscopy. Previously, five different nucleoid morphologies—square, crescent, rod, branched, and double rings—and their significance in different stages of cell growth have been shown in *D. radiodurans* (50). So, the mid-log-phase cells expressing drDivIVA-RFP were stained with SYTO-9 dye, and the nucleoid morphology and movement were monitored. These cells had a square- or early crescent-shaped nucleoid at $t = 0$, which changed sequentially from crescent to rod to branched and, finally, double ring shapes during different stages of cell division, and the positions of drDivIVA foci changed along the separation of the nucleoid (Fig. 7). For instance, drDivIVA was normally present at the cell periphery, where it localized along the central septum in cells preparing for genome segregation ($t = 30$ min). As the structure of the nucleoid started to change, drDivIVA seemed to change position along the direction of genome movement, which might signify the role of drDivIVA in genome segregation. A role of DivIVA in genome segregation has also been suggested in other bacteria (47). The effect of T19 phosphorylation on genome segregation was less apparent, possibly because of drDivIVA heterogeneity caused by the expression of both wild-type proteins chromosomally and T19 mutants episomally. Although the pattern of T19E localization with respect to genome segregation was the same as for the wild type, the T19E foci did move during genome separation (Fig. 7). The molecular basis of the T19E mutant's effect(s) on genome segregation is not known yet. Previously, drDivIVA interactions with cognate genome segregation proteins were demonstrated (42). So, the effect of phosphorylation's mimic in the form of T19E on drDivIVA interaction with the ParA2 (ParA of chromosome II) protein was monitored *in vivo*. The drDivIVA and T19A proteins showed interaction with ParA2, while the T19E protein did not interact with ParA2 *in vivo* (Fig. 8), suggesting that phosphorylation of drDivIVA at the T19 position may have negatively affected its interaction with genome segregation proteins.

## DISCUSSION

*Deinococcus radiodurans* is well characterized for its extraordinary resistance to DNA damage (25, 26). Like other cocci, it divides in alternate planes of cell division. However,

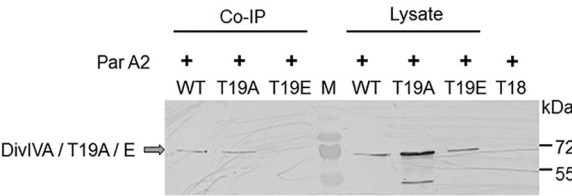

**FIG 8** The effect of T19 replacement on the interaction with chromosome II ParA (ParA2) *in vivo*. Deinococcal cells coharboring pVA2T18 (expressing ParA2 with the T18 tag) with pRGD4A (WT), pRGD4A^T19A (T19A), and pRGD4A^T19E (T19E) plasmids separately were checked for the expression of all three forms of the drDivIVA protein in cells. Cell lysates containing the same amounts of proteins were separated on 12% SDS-PAGE gels and immunoblotted with an anti-DivIVA antibody (Lysate). The T18 antibody was used to incubate cell extracts from these cells, and equal amounts of immunoprecipitates were separated on SDS-PAGE gels and probed using anti-DivIVA antibodies (Co-IP). Cells expressing the T18 tag alone on the pUT18 plasmid were used as a negative control.

unlike rod-shaped bacteria, such as *E. coli* and *B. subtilis*, there can be multiple virtual planes of cell division in cocci, and therefore, the determination of midcell position for the initiation of cell division can become a deterministic factor in the growth of round-shaped bacteria. Furthermore, gamma radiation induces growth arrest in this bacterium until the damaged DNA gets repaired (26). As it lacks a LexA/RecA-type canonical DNA damage response and cell cycle regulation (a paradigm in bacteria), the molecular basis of growth arrest would be an interesting mechanism to understand in this bacterium. Nevertheless, a DNA damage-responsive Ser/Thr quinoprotein kinase (RqkA) phosphorylates both DNA repair and cell division proteins in this bacterium (29, 39). Interestingly, phosphorylation of DNA repair proteins enhances their activity while negatively affecting the functions of the cell division proteins, such as FtsZ, *in vitro* (40, 51).

DivIVA, a member of the Min system family of proteins, localizes at the negative curvatures, including both the poles and the new division site, in *B. subtilis* cells and plays a crucial role in the spatial regulation of cell division (52). Furthermore, DivIVA localization at the negative curvatures is marked by cardiolipin microdomains in *B. subtilis* (52, 53). DivIVAs have been found to be the substrates of eukaryotic-like serine-threonine protein kinases (eSTPKs) in several bacteria, including *S. pneumoniae* and *Mycobacterium* spp. (9). The replacement of the phosphorylation site of the 201st threonine with alanine in DivIVA affects cell morphology, as marked by an elongated and bulged cell phenotype in *S. pneumoniae* (54). Notably, *D. radiodurans* lacks cardiolipin microdomains, and its DivIVA is an essential protein in the normal growth of this bacterium. Furthermore, it contains an extended C-terminal domain (CTD) compared with the DivIVA proteins of other bacteria, which contributes to the positioning of the septum during the normal growth of this bacterium (44). drDivIVA interacts with cognate cell division and genome segregation proteins through its N-terminal domain (42). In this study, we monitored the mechanisms underlying the localization of drDivIVA in the membrane and regulation of its function under both normal and gamma radiation-stressed growth conditions. We demonstrated that drDivIVA undergoes phosphorylation by RqkA at the T19 site, in close proximity with the membrane binding motif (previously known as the polarity-determining motif), and we investigated the *in vivo* effects of T19 phosphorylation on the protein's requirement in the cell division of this bacterium. We created phospho-mutant forms of this protein through site-directed mutagenesis to address whether phosphorylation affects its localization pattern and dynamics. The localizations of drDivIVA and its phospho-mutants and their dynamics with respect to FtsZ ring positioning and genome segregation were monitored. It was observed that at the initiation of cell division, the wild-type drDivIVA (mostly in dephosphorylated form) localized at the cell membrane and marked spatial territories for the Z-ring formation to take place in the zone that was free from drDivIVA (Fig. 9). During the next cycle of cell division, drDivIVA spread in its zone, and the foci in the mother cells were also observed in the daughter cells, which may possibly act as the memory foci for the cell to determine the plane of the next cell division, which would be perpendicular to the previous plane of division. The daughter cells also possessed foci at

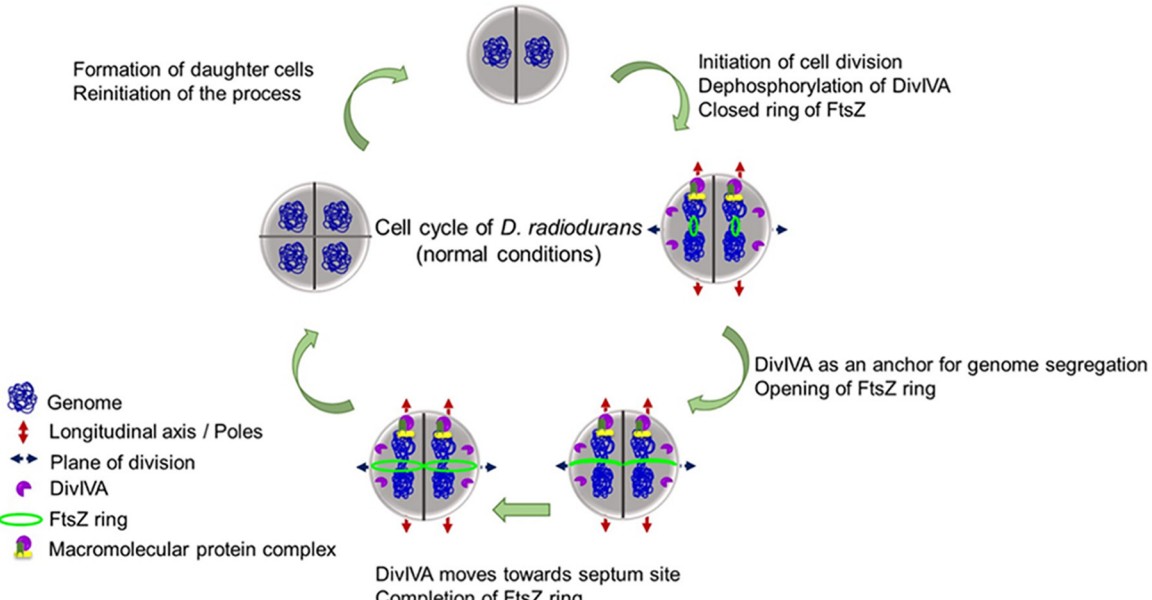

**FIG 9** A model depicting steps associated with the cell cycle of *D. radiodurans* under normal conditions. Schematic representation of the findings from this study that mostly fit into a model of cell division and genome segregation during normal growth of *D. radiodurans*.

the place where new cell division would occur. Thus, it was observed that drDivIVA marked both old and new planes of cell division in this coccus-shaped bacterium. In other work, DivIVA has been observed to mark the division site in *B. subtilis* and *S. aureus* (55, 56). The localization of drDivIVA as multiple foci at the membrane may appear random, but real-time monitoring by time-lapse confocal microscopy highlights that it plays a crucial role in spatial regulation in coherence with the positioning and dynamics of the FtsZ ring. The cellular positioning of drDivIVA was also analyzed with respect to the process of genome segregation, where it seemed to change position along the direction of genome segregation. The role of DivIVA in the process of genome segregation has been studied in other bacteria (55). These findings suggested that drDivIVA is a critical protein for cell division in this bacterium and that any disruption in its functional integrity can halt bacterial growth.

A possibility of T19 phosphorylation affecting the functional characteristics of drDivIVA was further investigated by creating phospho-ablative (T19A) and phospho-mimetic (T19E) mutants of this protein. It was observed that the behavior of T19A was highly similar to that of the wild-type protein during cell division. However, the mimicking of phosphorylation in the T19E mutant altered the typical localization and dynamics patterns of the wild-type protein in this bacterium. Time-lapse microscopic studies clearly showed that T19E did not show any dynamics during cell division. Since both the T19A and T19E proteins are phospho negative, and yet, they show different phenotypes, a strong possibility of the negative-charge component of phosphorylation contributing to the functional perturbation of the T19E mutant protein could be conjectured. The effect of a negative charge on the interaction of drDivIVA with the membrane was also investigated under a microscope, where cells expressing drDivIVA-RFP and its mutant versions were stained with the membrane binding dye vancomycin. The results largely ruled out the effect of a negative charge on membrane localization. However, a possible change in the interaction of the phospho-mimetic form of this protein with macromolecular assemblies that may be responsible for this protein's dynamics could not be ruled out, except that mimicking the phosphorylation in drDivIVA affected the wild-type interaction with genome-partitioning protein ParA2.

Since dr*divIVA* was found to be an essential gene, the functional integrity of T19E was also monitored by allelic replacement of the wild-type copy with both of the mutant alleles. Interestingly, we could replace the chromosomal copy of dr*divIVA* with *rfp*

fusions of dr*divIVA* (*divIVA-rfp*) and the phospho-ablative allele (*divIVA^T19A^-rfp*). Both of these derivatives showed similar growth patterns, as well as localizations and dynamics of the respective RFP fusions *in vivo*. But the homogenous replacement of wild-type copies with the *rfp* fusion of the phospho-mimetic allele (*divIVA^T19E^-rfp*) could not be achieved and the cells began to lose their viability. The replacement of the wild-type copy with the *divIVA^T19E^-rfp* allele was possible only in the presence of an episomal copy of drDivIVA. This further strengthened our arguments that the mimicking of phosphorylation at the T19 site in drDivIVA makes this protein nearly nonfunctional, which could not be tolerated by the bacterium for its growth even under normal conditions.

Since RqkA is a gamma radiation-responsive STPK and plays a significant role in the radioresistance of this bacterium, further insights into the effect of drDivIVA phosphorylation on gamma radiation resistance were investigated. For that, the localization patterns of wild-type drDivIVA and its phospho-mutants in the wild-type and *rqkA* mutant backgrounds of the *D. radiodurans* cells exposed to gamma radiation were studied. To our surprise, gamma radiation exposure had a distinct effect on the dynamics of wild-type drDivIVA in the presence and absence of RqkA in wild-type and *rqkA* mutant cells, respectively, while phospho-mutants in these genetic backgrounds showed almost no change in the localization patterns. For instance, when wild-type cells were exposed to gamma irradiation, the localization pattern of drDivIVA changed from multiple foci to fewer foci, whereas there was no change in the *rqkA* mutant. There are indications that the spread of drDivIVA-RFP foci under normal growth conditions seems to be due to the movement of protein from the original focal positions to new sites. Our results also showed that the number of RFP foci in wild-type cells expressing drDivIVA^T19E^-RFP episomally was lower than the number in wild-type cells expressing drDivIVA-RFP episomally (Fig. 2). Thus, these results indicate that phosphorylation of drDivIVA by RqkA in response to gamma radiation seems to have resulted in an arrest in the dynamics of this protein and thus, a lower number of foci per cell. The difference in the localization patterns of wild-type protein was observed in *rqkA*-plus cells (wild type) and *rqkA*-minus cells (*rqkA* mutant) after gamma radiation. On the other hand, the localization patterns drDivIVA-RFP and drDivIVA_T19E_-RFP were found to be under normal growth conditions. These results strongly suggests that RqkA phosphorylation of T19 has a negative effect on the dynamics of drDivIVA in *D. radiodurans*. Although further studies would be required to understand the mechanisms underlying the loss of drDivIVA dynamics upon T19 phosphorylation, as well as how the loss of drDivIVA function in response to gamma radiation affects the cell cycle, the available results clearly show that (i) RqkA phosphorylates drDivIVA at the T19 site, (ii) phosphorylation alters its *in vivo* dynamics, which may be required for its function in the determination of cell polarity and the plane of the next cell division, and (iii) phospho-mimicking at the T19 site makes this protein nonfunctional, as seen by the lack of complementation by the replacement of a wild-type copy under normal growth conditions. Since DivIVA's role in the determination of both cell polarity and alternate planes of cell division, as well as whether it contributes to cell cycle regulation in response to gamma radiation, has never been shown in any other bacteria but is shown here in the radioresistant bacterium *D. radiodurans*, this study signifies not just a demonstration of DivIVA phosphorylation by any STPK, rather than by a radiation-responsive STPK (RqkA), which could manifest the cell cycle arrest in this bacterium.

## MATERIALS AND METHODS

**Bacterial strains, plasmids, and bacterial growth measurement.** *D. radiodurans* R1 (ATCC 13939) was a kind gift from J. Ortner, Germany (57). The shuttle plasmids pVHS559 and pRADgro were suitably modified in our laboratory from the parental backbones of p11559 (58) and pRAD1 (59), respectively. The *E. coli* cells containing pVHS559 and its derivatives were maintained in the presence of spectinomycin at 40 $\mu$g/mL, while *D. radiodurans* was maintained in the presence of spectinomycin at 75 $\mu$g/mL, as described previously (60). Similarly, the pRADgro plasmid was maintained in *E. coli* with ampicillin (100 $\mu$g/mL) and in *D. radiodurans* with chloramphenicol (8 $\mu$g/mL) (61). Details of the plasmids and strains used in this study are listed in Table 1. The derived strains were grown in the presence of the

**TABLE 1** List of strains, primers, and plasmids used in the study

| Sl. no. | Name | Genotype, sequence, or description | Purpose or protein size (kDa) | Reference or source |
|---|---|---|---|---|
| **Strains** | | | | |
| 1 | *Deinococcus radiodurans* R1 | Wild-type ATCC 13939 | | 57 |
| 2 | *Escherichia coli* MG1655 | Wild-type | | Laboratory stock |
| 3 | *E. coli* DH5α | F⁻/endA1 hsdR17 glnV44 thi-1 recA1 gyrA relA Δ(lacIZYA-argF)U169 deoR [φ80dlacΔ(lacZ)M15] | | Laboratory stock |
| 4 | *E. coli* BL21 (DE3) pLysS | F⁻ ompT gal [dcm] [lon] hsdSB DE3::T7RNA | | Laboratory stock |
| 5 | *D. radiodurans* divIVA::divIVA-rfp | divIVA replaced with divIVA-rfp under its native promoter (Kan, 8 µg/mL) | | This study |
| 6 | *D. radiodurans* divIVA::divIVA^T19A-rfp | divIVA replaced with divIVA^T19A-rfp under its native promoter (Kan, 8 µg/mL) | | This study |
| 7 | *D. radiodurans* D4A^T19E + DivIVA | divIVA replaced with divIVA^T19E-rfp under its native promoter + pRGhis D4A (Kan, 8 µg/mL, and Cm, 5 µg/mL) | | This study |
| **Primers** | | | | |
| 1 | D4ADNFw | 5'-CGGGATCCAAACGGCCAGGCCGTCGT-3' | pNKUD4D | |
| 2 | D4ADNRw | 5'-GCTCTAGATCTCGGCCTGAGCACTGGC-3' | | |
| 3 | D4F | 5'-CGGGGCCCATGAGCTCGCCAATAAC-3' | Diagnostic PCR | |
| 4 | D4R | 5'-CGGAATTCTTACGAGCCGCTCACCCT-3' | | |
| 5 | D4CF | 5'-CGGGATCCATGAGCCTGAGCGGCAGTT-3' | | |
| 6 | D4R | 5'-CGGAATTCTTATTTCTCGTCGTCCAGC-3' | | |
| 7 | D4ADSFw | 5'-CGGGATCCGGCAGCTATGAGCTCGCCAATAAC-3' | pDsD4A | |
| 8 | D4ADSRw | 5'-GGGGTACCGCGGCCGCTTTCTCGTCGTCCAGC-3' | | |
| 9 | D4ARFPFw | 5'-CGGAGCTCGGGCCCCACCACCACCACCATGCCTGCAGGTCG-3' | pNKD4A | |
| 10 | D4ARFPRw | 5'-CGGATATCTAGACTGAGGCCGCTACAG-3' | | |
| 11 | RFPRw | 5'-CGGAATTCCTACAGGAACAGGTGGTGGCG-3' | pRGD4A | |
| 12 | HisRedFw | 5'-CGGAGCTCGGGCCCCACCACCACCACCACCATGCCTGCAGGTCG-3' | pRGD4A^T19A, pRGD4A^T19E | |
| 13 | RedRw | 5'-CGGATATCTAGACTGAGGCCGCTACAG-3' | Diagnostic PCR | |
| 14 | nptIIFw | 5'-GCACGGTGCCGAGTGG-3' | | |
| 15 | nptIIRw | 5'-GTCAGCGTAATGCTCTG-3' | | |
| **Plasmids** | | | | |
| 1 | pNOKOUT | pBSK⁺ (Amp^r) containing nptII cassette (937 bp) at SmaI site, ~4.0 kb | | 62 |
| 2 | pRADgro | pRAD1 containing groESL promoter (261 bp) from *D. radiodurans* at BglII-XbaI site, ~6.5 kb, Amp^r | | 61 |
| 3 | pRGhisD4A | pRADgro containing his-divIVA at ApaI-XbaI site, ~7.7 kb, Amp^r | ~40 | This study |
| 4 | pRGD4A | pRADgro containing his-divIVA-rfp at ApaI-XbaI site, ~8.2 kb, Amp^r | ~69 | This study |
| 5 | pRGD4A^T19A | pRADgro containing his-divIVA^T19A-rfp at ApaI-XbaI site, ~8.2 kb, Amp^r | ~69 | This study |
| 6 | pRGD4A^T19E | pRADgro containing his-divIVA^T19E-rfp at ApaI-XbaI site, ~8.2 kb, Amp^r | ~69 | This study |
| 7 | pVHZGFP | pVHSM containing ftsZ-gfp at SacI-AflII site, ~11.7 kb, Spec^r | ~61 | 27 |
| 8 | pDsRed | ~3.3-kb plasmid; N-terminal red fluorescent tag (Amp^r) | | Clontech, Inc. |
| 9 | pDsD4A | pDsRed + divIVA at BamHI and KpnI, ~4.3 kb, Amp^r | ~69 | This study |
| 10 | pDsD4A^T19A | pDsRed + divIVA^T19A at BamHI and KpnI, ~4.3 kb, Amp^r | ~69 | This study |
| 11 | pDsD4A^T19E | pDsRed + divIVA^T19E at BamHI and KpnI, ~4.3 kb, Amp^r | ~69 | This study |
| 12 | pNKD4A | pNOKOUT containing divIVA-rfp (~1.7 kb) and downstream (~1.0 kb) sequences of divIVA, ~6.7 kb, Amp^r/Kan^r | This study | This study |
| 13 | pNKD4A^T19A | pNOKOUT containing divIVA^T19A-rfp (~1.7 kb) and downstream (~1.0 kb) sequences of divIVA, ~6.7 kb, Amp^r/Kan^r | This study | This study |
| 14 | pNKD4A^T19E | pNOKOUT containing divIVA^T19E-rfp (~1.7 kb) and downstream (~1.0 kb) sequences of divIVA, ~6.7 kb, Amp^r/Kan^r | This study | This study |

respective antibiotics, and their growth at an optical density of 600 nm ($OD_{600}$) was monitored in a sterile 24-well microtiter plate using a Synergy H1 hybrid multimode microplate reader (BioTek) overnight at 32°C. The data were processed, analyzed for statistical significance, and plotted using GraphPad Prism software.

**Protein phosphorylation and immunoblotting.** Recombinant drDivIVA and RqkA were expressed on pETD4A (43) and pET2518 (39) plasmids, respectively, in transgenic *E. coli* BL21 cells and purified using protocols described earlier. The recombinant drDivIVA was incubated for 30 min at 37°C with different molar ratios (X/100, X/50, X/25, and X/10, where X is DivIVA) of recombinant RqkA in a kinase reaction mixture containing 10 mM Tris-HCl (pH 7.6), 20 mM KCl, 1.5 mM $MgCl_2$, 0.5 mM dithiothreitol, 2% glycerol, 0.1 mM EDTA, and 5 mM unlabeled (cold) ATP and [$\gamma$-$^{32}$P]ATP. The reaction was stopped by adding 2× SDS sample buffer, and the phosphorylation was checked by 12% SDS-PAGE as described previously (43). The gel was dried and exposed to X-ray film, and autoradiograms were developed. The data were further analyzed and plotted using GraphPad Prism software.

The phosphorylation of drDivIVA by RqkA was further checked *ex vivo* in *E. coli* BL21 cells coexpressing drDivIVA and RqkA episomally. The coexpression of both proteins was ascertained independently using specific antibodies. The recombinant drDivIVA protein was purified from the *E. coli* cells coexpressing RqkA on the plasmid and checked for phosphorylation using phospho-Ser/Thr epitope antibodies as described previously (39). In brief, purified drDivIVA from *E. coli* expressing RqkA (P-DivIVA) and the control (DivIVA) were loaded on a 12% SDS-PAGE gel and transferred to a polyvinylidene difluoride (PVDF) membrane. Polyclonal phospho-Ser/Thr epitope antibodies were used to hybridize the blots, and signals were detected using alkaline phosphatase-conjugated anti-rabbit secondary antibodies. Similarly, the phosphorylation status of drDivIVA from *D. radiodurans* cells (both unirradiated [UI] and irradiated [I]) was also checked by immunoblotting. The cell lysates of *D. radiodurans* expressing pRGhisD4A were loaded on a 12% SDS-PAGE gel and transferred to a PVDF membrane, and the blots were hybridized with both anti-DivIVA and polyclonal phospho-Ser/Thr epitope antibodies.

**Mapping of phosphorylation sites in drDivIVA.** Phosphorylated drDivIVA (coexpressed with recombinant RqkA) in *E. coli* and the control protein were purified to homogeneity. Proteins were separated on a SDS-PAGE gel, followed by staining with Coomassie brilliant blue. The drDivIVA protein bands were incised from the gel and reduced with 5 mM TCEP [Tris(2-carboxyethyl)phosphine hydrochloride], followed by alkylation with 50 mM iodoacetamide before being digested with 1 $\mu$g trypsin at 37°C for 16 h. The peptide mixture was purified using a C18 column (The Nest Group, Southborough, MA) according to the manufacturer's protocol and dried under vacuum. The peptide pellet was dissolved in buffer A (5% acetonitrile/0.1% formic acid), and the mixture was resolved using a 15-cm PicoFrit column (360-um outer diameter, 75-um inner diameter, 10-um tip) filled with 1.8 $\mu$m C18 resin (Dr. Maisch) in an Easy-nLC 1000 system (Thermo Fisher Scientific) coupled to a Q-Exactive mass spectrometer (Thermo Fisher Scientific), by outsourcing. In brief, the peptides were loaded with buffer A and eluted with a 0 to 40% gradient of buffer B (95% acetonitrile/0.1% formic acid) at a flow rate of 300 nl/min for 40 min. The Q-Exactive was operated using the Top10 HCD data-dependent acquisition mode with a full scan resolution of 70,000 at *m/z* 400. Tandem mass spectrometry (MS/MS) scans were acquired at a resolution of 17,500 at *m/z* 400. The lock mass option was enabled for polydimethylcyclosiloxane (PCM) ions (*m/z* = 445.120025) for internal recalibration during the run. The data were processed against the Uniprot *D. radiodurans* database using proteome discoverer software, with phosphorylation as the dynamic modification criterion.

**Site-directed mutagenesis.** Site-directed mutagenesis was carried out using the Q5 site-directed mutagenesis kit, following the manufacturer's protocol (NEB), including the primers carrying the altered codon, which were designed using the online tool OligoCalc. In brief, the primers containing the desired mutation in the threonine codon to change it into either an alanine or glutamate codon were used for amplification of the plasmid carrying *divIVA* (pUTD4A) using the standard procedure. PCR products were incubated with an enzyme mixture containing a kinase, a ligase, and DpnI at 25°C for 30 min. Together, these enzymes allowed rapid circularization of the PCR product and the removal of the nonmutated DNA template by DpnI. The circularized plasmid was transformed into *E. coli* NovaBlue cells, and mutations were ascertained by sequencing. The phospho-mutant versions of drDivIVA were cloned separately into different expression plasmids.

**Construction of recombinant plasmids.** The recombinant plasmids pDsD4A and pRGD4A were constructed for the monitoring of the localization of drDivIVA (DR_1369) as DivIVA-RFP (43). Similarly, plasmids pDsD4A$^{T19A}$ and pRGD4A$^{T19A}$ and plasmids pDsD4A$^{T19E}$ and pRGD4A$^{T19E}$ were constructed for the expression of T19A-RFP and T19E-RFP, respectively. In brief, the coding sequences of DivIVA and its T19A and T19E phospho-mutant alleles were PCR amplified using sequence-specific primers (Table 1) and cloned at BamHI-KpnI sites in the pDsRed plasmid (Clontech, Inc.), yielding pDsD4A$^{T19A}$ and pDsD4A$^{T19E}$, respectively. Further, the translation fusions *divIVA*$^{T19A}$-*rfp* and *divIVA*$^{T19E}$-*rfp* were amplified from pDsD4A$^{T19A}$ and pDsD4A$^{T19E}$, respectively, using RedFw and RedRw primers, and cloned in pRADgro to yield pRGD4A$^{T19A}$ and pRGD4A$^{T19E}$, respectively. The recombinant plasmids pNKD4A, pNKD4A$^{T19A}$, and pNKD4A$^{T19E}$ were constructed to replace the wild-type *divIVA* allele with *divIVA-rfp*, *divIVA*$^{T19A}$-*rfp*, and *divIVA*$^{T19E}$-*rfp* alleles in the chromosome for the expression of DivIVA-RFP, T19A-RFP, and T19E-RFP, respectively, under the native promoter in *D. radiodurans*. For that, *divIVA-rfp* alleles and the downstream region of the coding sequence of drDivIVA were separately PCR amplified using sequence-specific primers (Table 1). *divIVA-rfp* and its phospho-mutant alleles were cloned at ApaI-EcoRV, while the downstream sequences of DivIVA were cloned at the BamHI-XbaI sites in pNOKOUT (Kan$^r$), to give pNKD4A, pNKD4A$^{T19A}$, and pNKD4A$^{T19E}$, respectively.

**Generation of chromosomal insertions in *divIVA*.** For generating chromosomal insertions, the *D. radiodurans* cells were transformed with XmnI-linearized pNKD4A, pNKD4A$^{T19A}$, and pNKD4A$^{T19E}$

plasmids using protocols described previously (62). The transformants were grown for several rounds in the presence of kanamycin (8 $\mu$g/mL). The cycles of alternate streaking on tryptone-yeast extract-glucose (TYG) agar plates and subculturing in TYG broth were repeated for several rounds until homogenous replacement of the target with insertion sequences was achieved (44). Genomic DNA was prepared, and homogenous replacement of *divIVA* with the expression cassettes of *divIVA-rfp-nptII* and other phospho-mutant alleles through homologous recombination was ascertained by diagnostic PCR (see Fig. S2 at https://barc.gov.in/publications/mbio/DivIVA/figures_s.pdf) with gene-specific primers as given in Table 1. A 1% agarose gel was used to examine the PCR products. The replacement of the wild-type allele with the T19E allele could not be obtained as the cells did not survive. When drDivIVA was expressed episomally on the pRGhisD4A plasmid, a *divIVA*$^{T19E}$-*rfp* insertion was created in place of the *divIVA* allele. The confirmation of insertions of *rfp-nptII* downstream from *divIVA* in the correct reading frame was monitored by fluorescence microscopy examining the expression of drDivIVA-RFP and its variants under the native promoter.

**Fluorescence microscopy and image analysis.** The general procedures of fluorescence microscopy using a confocal microscope (Olympus IX83 FluoView 3000) were followed as described previously (44). In brief, the bacterial cultures were grown in TYG broth, fixed with 4% paraformaldehyde for 10 min on ice, and washed twice with phosphate-buffered saline (pH 7.4). These cells were stained with DAPI (4′,6-diamidino-2-phenylindole; 0.5 $\mu$g/$\mu$L) for 10 min on ice and then washed thrice with phosphate-buffered saline (PBS). The cells were resuspended in PBS and mounted on a 1% agarose bed on glass slides, and samples were observed under the microscope. For FtsZ localization, the cells expressing FtsZ-GFP on the pVHZGFP plasmid (27) were grown overnight and cells equivalent to an OD$_{600}$ of 0.05 to 0.1 were diluted with fresh medium containing the required antibiotic(s) and induced with 10 mM IPTG (isopropyl $\beta$-D-thiogalactopyranoside) overnight. For time-lapse imaging, the exponentially growing cells were suspended in PBS and mounted on an agarose pad made of 2× TGY and designed with air holes to oxygenate the cells. Confocal microscopy was carried out using laser beams focused to the back focal plane of a 100× 1.40 numeric aperture oil-immersion apochromatic objective lens (Olympus). The laser parameters for illumination at the sample were tuned using the installed FluoView software. For imaging at the required time points, a series of Z-planes were acquired every 400 nm using the motorized platform. Fluorescence emissions were collected through a DM-405/488/561 dichroic mirror and the corresponding single-band emission filters. For constructing three-dimensional (3-D) images, the cells expressing FtsZ-GFP and drDivIVA-RFP or its phospho-mutants were sliced in Z-planes every 400 nm at 45- to 60-min intervals for a period of 4 to 5 h using very-low-power lasers at 561 nm and 488 nm. To monitor the movement of the genome, the cells expressing drDivIVA-RFP and its phospho-mutants were stained with 150 nM SYTO-9 (Syto Green) dye, and images were acquired for a period of 2 to 3 h after every 30 to 45 min using very-low-power lasers at 561 nm and 488 nm. For image analysis, images of cells captured from at least two separate microscopic fields in two independent experiments were analyzed for the required attributes. The automated Cell Sens software was used to perform the image analysis and to measure other cell parameters. Data obtained were subjected to Student's *t* test analysis using statistical programs in GraphPad Prism software.

The cells in exponential phase were exposed to a 6-kGy dose of gamma radiation for imaging of drDivIVA and its phospho-mutant forms in both wild-type and Δ*rqkA* mutant backgrounds, while control cells were kept on ice. Both unirradiated (UI) and irradiated (I) cells expressing DivIVA$^{WT}$, DivIVA$^{T19A}$, and DivIVA$^{T19E}$ were collected at 0 h and 2 h of postirradiation recovery (PIR) after gamma irradiation. The cells were processed for confocal microscopy as described above. Approximately 250 to 300 cells were analyzed for the required attributes and processed by using Cell Sens and Adobe Photoshop 7.0 software.

**Coimmunoprecipitation (co-IP) of drDivIVA and its phospho-mutant forms with ParA2.** *D. radiodurans* R1 cells expressing drDivIVA-RFP (wild-type and phospho-mutant forms) were transformed with ParA2-T18 (pVT18A2) (63), and colonies on spectinomycin (70 $\mu$g/mL) were scored. These colonies were further grown and induced with 20 mM IPTG overnight. The cell extracts of *D. radiodurans* expressing these constructs were prepared along with those of the vector control. In brief, deinococcal cells expressing the desired proteins were pelleted and washed with 70% ethanol, followed by a wash with PBS. Pellets were suspended in 500 $\mu$L of lysis buffer A (50 mM Tris, pH 7.5, 100 mM NaCl, 1 mM phenylmethylsulfonyl fluoride [PMSF], 5 mM MgCl$_2$, 1 mM dithiothreitol, 0.5% Triton X-100) with 0.5 mg/mL lysozyme and 50 $\mu$g protease inhibitor cocktail tablet (Roche Biochemicals), followed by sonication on ice. Cell debris was removed by centrifugation at 2,000 × *g* for 10 min at 4°C. Monoclonal antibodies against the T18 tag were used to immunoprecipitate the clear cell extracts according to the protein G immunoprecipitation kit (Merck, Inc.) protocol. Immunoprecipitates were separated on 10% SDS-PAGE gels, blotted onto PVDF membranes, and hybridized with anti-DivIVA antibody. Hybridization signals were detected using anti-mouse secondary antibodies conjugated with alkaline phosphatase using BCIP/NBT (5-bromo-4-chloro-3-indolylphosphate/nitroblue tetrazolium) substrates (Merck, Inc.).

**Data availability.** This study includes no data deposited in external repositories.

## ACKNOWLEDGMENTS

We thank Sebastian Raja for his help in confocal microscopy and Bhakti Basu, Anand Ballal, and Ganesh K. Maurya for their critical reading of the manuscript and comments.

Reema Chaudhary is grateful to the Department of Atomic Energy, India, for her research fellowship.

We have no relevant financial or nonfinancial interests to disclose.

R.C., experiments, data analysis, discussion and manuscript writing; S.K., experiments, data analysis, discussion and manuscript writing; H.S.M., principal investigator, conceptualization of

idea, data analysis, discussion, manuscript writing, and communication for publication. The authors all agreed to this publication.

The manuscript has been checked for plagiarism and similarities by our institutional committee.

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
