## [Reviewer comments · Microbiology Spectrum]

Microbiology Spectrum

DivIVA phosphorylation affects its dynamics and cell cycle in radioresistant *Deinococcus radiodurans*

Reema Chaudhary, Swathi kota, and Hari Misra

Corresponding Author(s): Hari Misra, Bhabha Atomic Research Centre

Review Timeline:

Submission Date:	August 10, 2022
Editorial Decision:	October 1, 2022
Revision Received:	November 30, 2022
Accepted:	January 6, 2023

Editor: Varsha Singh

Reviewer(s): Disclosure of reviewer identity is with reference to reviewer comments included in decision letter(s). The following individuals involved in review of your submission have agreed to reveal their identity: Dagmara Jakimowicz (Reviewer #2)

Transaction Report:

DOI: <https://doi.org/10.1128/spectrum.03141-22>

September 25, 2022

Prof. Hari S. Misra
Bhabha Atomic Research Centre
47 Shreeniketan
Anushaktinagar
Mumbai, N/A 400094
India

Re: Spectrum03141-22 (DivIVA phosphorylation affects its dynamics and cell cycle in radioresistant *Deinococcus radiodurans*)

Dear Prof. Hari S. Misra:

Thank you for submitting your manuscript to Microbiology Spectrum. We have now received comments on your manuscript. Both reviewers agree that the manuscript intends to address an interesting question of interest to microbiologists as well as cell division enthusiasts. However, one of the reviewers, an expert in the field, has raised major concern about the work. We can consider a substantially revised manuscript where the reviewers' comments are addressed by the inclusion of additional experiments. We would also recommend careful editing of the text and toning down statements which may overstate findings.

When submitting the revised version of your paper, please provide (1) point-by-point responses to the issues raised by the reviewers as file type "Response to Reviewers," not in your cover letter, and (2) a PDF file that indicates the changes from the original submission (by highlighting or underlining the changes) as file type "Marked Up Manuscript - For Review Only". Please use this link to submit your revised manuscript - we strongly recommend that you submit your paper within the next 90 days or reach out to me. Detailed instructions on submitting your revised paper are below.

Link Not Available

Sincerely,

Varsha Singh

Journals Department
Reviewer comments:

Reviewer #1 (Comments for the Author):

The authors present a fascinating study elucidating the effects of phosphorylated DivIVA on *D. radiodurans*. They present a well-designed method that led to novel data helping to understand the role of this protein in response to radiation. This work represents an important piece of science since *D. radiodurans* is also a model for cell cycle and DNA damage response in bacteria. I suggest working on the following concerns to improve the manuscript:

Abstract: This is a summary of the results. It is not clear what information is coming from the background and what is the result of this work. The authors should present an organized and informative abstract with an introduction, objective, methods, and main results/conclusions

Line 76: the authors stated, "for the first time, we report the phosphorylation of drDivIVA by RqkA", but they aforementioned it was known on Ref 25.

Line 221: I suggest avoiding not shown data results. The authors can show evidence as supplementary material

Fig1. Improve organization. The letters are too large and the labels are too close to one another. Is hard to understand. What are the authors showing with the SDS-Page on C and D? also in D, the SDS-PAGE does not seem to match with the first blot (P-DivIVA). I strongly suggest using a better image for P-DivIVA(D) since lanes are shifted from the other images

Fig8 lacks quality. The background is kind of scratched, the A panel indicates MW, but we cannot see it as in panel B. There is not indicated in the legend what is panel A and Panel B. I could guess A is the complete lysate and B is after the immunoprecipitation with Tag T8. Or, are those just repetitions? Please clarify. Why is T8 control in A, but not in B? Why are there two bands on Pane IA DivIVA-T19A? A control staining of ParA2 would be an interesting loading control for better interpretation of the results.

402-412: References to these ideas are required.

Discussion from 402-431 seems like an introduction again. The authors should focus on clearly establishing the state of the art regarding the association RqkA-DivIVA, and the advances they present in this work for the understanding of the effects of pDivIVA. As well as compare these functions with known interactions in other species.

Authors should carefully check the overestimation of their results in the discussion. For example statements in 440-442 and 434-436 were not proven or tested in this work and might be subjective appreciations from the authors.

Other minor comments:

Line 23: typo in "o"

Line30: Italics on "in vivo", the error is also repeated in other sections

Line 60: italics on bacterium name

Fig2C and E: indicate how many cells were counted for the statistical analysis

Reviewer #3 (Comments for the Author):

The work by Chaudhari et al reports a series of experiments trying to understand the role of DivIVA phosphorylation in the radiation-resistant bacterium *Deinococcus radiourans*.

The authors present experiments designed to show that DivIVA is phosphorylated by the protein kinase RqkA at residue threonine 19. They go on and investigate the effect of phosphor-ablative and -mimetic amino acid exchanges on DivIVA localization. Apparently DivIVA T19E is altered with respect to its localization inside *D. radiourans* cells, while T19A is not. The effect of both mutations was also studied and according to their experiments T19E was interpreted as a lethal mutation. Finally, chromosome segregation was studied in cells expressing divIVA T19 variants, but a clear-cut result was not generated. Lastly, an immunoprecipitation experiment is shown that should suggest that DivIVA T19E does not longer interact with ParA.

I generally find the research question raised in this manuscript interesting enough to be considered for publication by a general microbiology journal with an above-average impact factor. However, I have several general and serious concerns regarding the validity of the conclusions that are often way too speculative and not supported by the data. Moreover, certain experiments are not well-described or their design excludes the intended answer right from the beginning. In several cases, important controls are missing. Due to its often hyper-speculative nature, the manuscript is not easy to follow. I do not see this manuscript ready for publication in any scientific journal in its present form, therefore I only will comment on the results part

Specific remarks:

Fig. 1A+B: Coomassie stained gels are missing that confirm equal gel loading of the radiograms.

Lane 235: PIR - unexplained abbreviation

Fig. S1A-B: It is unclear, which of the peaks correspond to the phosphorylated residue, a comparison of the raw data signals for p-DivIVA and unphosphorylated DivIVA is missing to convince the reader that DivIVA is phosphorylated at T19 (and not phosphorylated in unphosphorylated DivIVA or DivIVA T19A).

Lane 248: At least since the clarification of the DivIVA crystal structure, the "polar determining motif" is and outdated concept.

The mentioned residues are for membrane binding. Important literature describing this is not cited (see PMIDs: 30887576, 20502438). Moreover, according to the alignment shown in Fig. S1C, T19 is not part of the "polar determining motif".

Fig. S1E: positive control is missing

Lane 279 onwards: T19E is less expressed than wt DivIVA - I would suspect that this mutation (introduction of a negative charge) alters the structure of the N-terminal lipid binding domain. As a simple test that at least membrane binding and the formation of higher order aggregates are not compromised by T19E, the authors should test oligomerization of wt and T19E DivIVA, for example by blue native PAGE, and membrane binding by cell fractionation (membrane vs cytosolic extracts) with subsequent Western blotting to determine the distribution of DivIVA between the membrane and the cytosol.

Lane 295: The claim that "drDivIVA protein localizes in two planes, (i) in the plane of the FtsZ ring and (ii) in the direction of genome segregation" is an overinterpretation and is not supported by the images shown. Moreover, the statement that "this suggests the possible role of this protein in both cell division and genome segregation by an unknown mechanism" is mere speculation. How can a spotty drDivIVA localization pattern suggest this? To me, the localization pattern seems rather uninformative.

Lane 302-304: As a role of drDivIVA, its localization and phosphorylation in chromosome segregation and perpendicularity of cell division has not conclusively been shown, this sentence is mere overinterpretation.

Lane 309: The chosen citation (43) refers to a review and not to original work. Please cite the work that shows that drdivIVA can only be deleted in the presence of a second copy.

Lane 318-321: I find the way to demonstrate essentiality of divIVAT19E not convincing. The authors should construct a mutant with divIVAT19E under control of a constitutive promoter in addition to a wt copy of divIVA under control of an inducible promoter and test this strain for inducer dependency.

Fig. 4D: DivIVA signals in the first two and last three lanes have different molecular weights. Why?

Lane 333: The claim that Fig. S2 shows time-lapse data is not true, as different cells at different time points are shown (as stated in the the figure legend).

Lane 341-343: blue and black arrows are invisible in Fig. 5.

Lane 343-345: Highly speculative and not supported by data.

Lane 367-368: I do not see the claim that "the pattern of DivIVA localization has changed..." supported by the microscopic images. I am also not convinced that the quantified data shown in Fig. 6B for wt after irradiation (clear effect) correctly represent the corresponding raw data shown in Fig. 6A (no clear effect). I also have the impression from the data in Fig. 6A-B that DivIVAT19A localization is different from wtDivIVA (more foci), which is in contrast to their earlier claim that DivIVAT19A localizes as wtDivIVA (lanes 347-352).

Lane 388-389: "As the structure of the nucleoid changes...genome movement" - Mere overinterpretation, neither "DivIVA movement" nor "DivIVA movement along the direction of chromosome movement" is shown.

Lane 392-393: "Although the pattern of T19E localization ...was evidenced". Again, overinterpretation of data.

Staff Comments:

Preparing Revision Guidelines

Please return the manuscript within 60 days; if you cannot complete the modification within this time period, please contact me. If you do not wish to modify the manuscript and prefer to submit it to another journal, please notify me of your decision immediately so that the manuscript may be formally withdrawn from consideration by Microbiology Spectrum.

Corresponding authors may join or renew ASM membership to obtain discounts on publication fees. Need to upgrade your

membership level? Please contact Customer Service at Service@asmusa.org.

Authors' response to reviewers' comments

Title of the research article: DivIVA phosphorylation affects its dynamics and cell cycle in radioresistant *Deinococcus radiodurans*. [Paper #Spectrum03141-22]

Editors' comments and suggestions:

Thank you for submitting your manuscript to Microbiology Spectrum. We have now received comments on your manuscript. Both reviewers agree that the manuscript intends to address an interesting question of interest to microbiologists as well as cell division enthusiasts. However, one of the reviewers, an expert in the field, has raised major concern about the work. We can consider a substantially revised manuscript where the reviewers' comments are addressed by the inclusion of additional experiments. We would also recommend careful editing of the text and toning down statements which may overstate findings.

Authors' response: *We thank you for getting this work reviewed by the experts in the field and your decision for revision. We have revised it by addressing almost all the concerns of the reviewers with some new experiments. Hope you find it suitable for publication in this journal.*

When submitting the revised version of your paper, please provide

- (1) point-by-point responses to the issues raised by the reviewers as file type "Response to Reviewers," not in your cover letter, and

Authors' response: *Thank you. It has been provided.*

- (2) A PDF file that indicates the changes from the original submission (by highlighting or underlining the changes) as file type "Marked Up Manuscript - For Review Only". Please use this link to submit your revised manuscript - we strongly recommend that you submit your paper within the next 90 days or reach out to me. Detailed instructions on submitting your revised paper are below.

Authors' response: *Thank you. It has been provided.*

Reviewer comments:

Reviewer #1 (Comments for the Author):

The authors present a fascinating study elucidating the effects of phosphorylated DivIVA on *D. radiodurans*. They present a well-designed method that led to novel data helping to understand the role of this protein in response to radiation. This work represents an important piece of science since *D. radiodurans* is also a model for cell cycle and DNA damage response in bacteria. I suggest working on the following concerns to improve the manuscript:

Authors' response: *Thank you so much for appreciating our efforts and your encouraging words.*

Abstract: This is a summary of the results. It is not clear what information is coming from the background and what is the result of this work. The authors should present an organized and informative abstract with an introduction, objective, methods, and main results/conclusions.

Authors' response: *Thank you. Abstract has been modified as per suggestions and within the acceptable format of the journal.*

Line 76: the authors stated, "For the first time, we report the phosphorylation of drDivIVA by RqkA", but they aforementioned it was known on Ref 25.

Authors' response: *Thank you so much for comment. Ref. 25 has no mention of phosphorylation in drDivIVA but for some other cell division proteins of D. radiodurans. Sorry for this mistake and the sentence has been modified.*

Line 221: I suggest avoiding not shown data results. The authors can show evidence as supplementary material.

Authors' response: *Thank you. We have added the data as supplementary figure (Figure S1) showing the expression of RqkA from the plasmid pRadRqkA in the cells harboring pETD4A.*

Fig1. Improve organization. The letters are too large and the labels are too close to one another. Is hard to understand. What are the authors showing with the SDS-Page on C and D? Also in D, the SDS-PAGE does not seem to match with the first blot (P-DivIVA). I strongly suggest using a better image for P-DivIVA (D) since lanes are shifted from the other images.

Authors' response: *Thank you for your suggestion. Figure 1 has been organized in the revision as per the suggestions. We have shown the SDS-PAGE images just to highlight the uniform loading of our lysates. The blot showing P-DivIVA is changed in the D panel to address the raised concern.*

Fig8 lacks quality. The background is kind of scratched, the A panel indicates MW, but we cannot see it as in panel B. There is not indicated in the legend what is panel A and Panel B. I could guess A is the complete lysate and B is after the immunoprecipitation with Tag T8. Or, are those just repetitions? Please clarify. Why is T8 control in A, but not in B? Why are there two bands on Pane 1A DivIVA-T19A? A control staining of ParA2 would be an interesting loading control for better interpretation of the results.

Authors' response: *Thank you for comment. Actually, both the panels of Fig 8 were cropped from the same blot to make it easy to understand. Now, it has been provided in the original uncropped form. We loaded equal amount of protein by estimation. I agree, A2 blotting should have been a better control but that was not recorded. Also, because the amount of immunoprecipitate was not enough for running another SDS-PAGE gel for CBB staining, the SDS PAGE gel the one used for blotting was stained after blotting and given below for review purpose. Legend has been made clearer.*

402-412: References to these ideas are required.

Authors' response: *Thank you. The required references have been added.*

Discussion from 402-431 seems like an introduction again. The authors should focus on clearly establishing the state of the art regarding the association RqkA-DivIVA, and the advances they present in this work for the understanding of the effects of pDivIVA. As well as compare these functions with known interactions in other species.

Authors' response: Sorry for the inconvenience. Discussion has been modified for the better clarity.

Authors should carefully check the overestimation of their results in the discussion. For example statements in 440-442 and 434-436 were not proven or tested in this work and might be subjective appreciations from the authors.

Authors' response: Thank you. The statements have been modified in the revision.

Other minor comments:

Line 23: typo in "o"

Authors' response: Sorry for the typo error. It has been corrected.

Line30: Italics on "in vivo", the error is also repeated in other sections.

Authors' response: It has been corrected.

Line 60: italics on bacterium name.

Authors' response: It has been corrected.

Fig2C and E: indicate how many cells were counted for the statistical analysis.

Authors' response: Thank you. It has been mentioned in the materials and methods section that 250-300 cells were analyzed for the required attributes.

Reviewer #3 (Comments for the Author):

The work by Chaudhari et al reports a series of experiments trying to understand the role of DivIVA phosphorylation in the radiation-resistant bacterium *Deinococcus radiourans*. The authors present experiments designed to show that DivIVA is phosphorylated by the protein kinase RqkA at residue threonine 19. They go on and investigate the effect of phosphor-ablative and -mimetic amino acid exchanges on DivIVA localization. Apparently DivIVA T19E is altered with respect to its localization inside *D. radiourans* cells, while T19A is not. The effect of both mutations was also studied and according to their experiments T19E was interpreted as a lethal mutation. Finally, chromosome segregation was studied in cells expressing divIVA T19 variants, but a clear-cut result was not generated. Lastly, an immunoprecipitation experiment is shown that should suggest that DivIVA T19E does not longer interact with ParA. I generally find the research question raised in this manuscript interesting enough to be considered for publication by a general microbiology journal with an above-average impact factor.

Authors' response: Thank you so much for reviewing our manuscript, appreciating work with the constructive criticisms.

However, I have several general and serious concerns regarding the validity of the conclusions that are often way too speculative and not supported by the data. Moreover, certain experiments are not well-described or their design excludes the intended answer right from the beginning. In several cases, important controls are missing. Due to its often hyper-speculative nature, the manuscript is not easy to follow. I do not see this manuscript ready for publication in any scientific journal in its present form, therefore I only will comment on the results part

Authors' response: We thank you for your constructive comments that have helped us to further improve our manuscript.

Specific remarks:

Fig. 1A+B: Coomassie stained gels are missing that confirm equal gel loading of the radiograms.

Authors' response: Thank you. The SDS-PAGE data is shown below for quick reference and included in revised figure.

Lane 235: PIR - unexplained abbreviation

Authors' response: Thanks. PIR is abbreviation of 'Post-Irradiation Recovery' and used in context when the cells are irradiated and allowed to recover during post-irradiation. This has been corrected.

Fig. S1A-B: It is unclear, which of the peaks correspond to the phosphorylated residue, a comparison of the raw data signals for p-DivIVA and unphosphorylated DivIVA is missing to convince the reader that DivIVA is phosphorylated at T19 (and not phosphorylated in unphosphorylated DivIVA or DivIVA T19A).

Authors' response: Thank you for your suggestion. The chromatogram showing the phosphorylated peak is shown below for the reference (C) and has been included in revised figure. Though, the already presented chromatogram shows only the identity of the protein as DivIVA of this bacterium. This figure has been modified accordingly.

Fig S1: Phospho-site mapping in deinococcal DivIVA. The phosphorylated DivIVA protein was subjected to mass-spectrometry for identification of the phospho-site. A chromatogram of the protein was generated and ion-series of the phospho-peptide fragment were also analyzed with respect to non-phospho form of the protein (A-B). The chromatogram has highlighted the phospho-peptide peaks (C). Pair-wise alignment of the first few amino acids of DivIVA of *B. subtilis* and *D. radiodurans* was highlighted to show the proximity of phosphorylation and polar motifs (D). Phospho-peptide was found to be DI*T*HQSFDGR where T is mutated to alanine (A; phospho-ablative) and glutamate (E; phospho-mimetic) by site-directed mutagenesis (E). The mutant forms were also checked for phosphorylation *ex-vivo*. *E. coli* cells harbouring pRadRqkA and any one of the following plasmids, pRGD4A (DivIVA^{WT}), pRGD4A^{T19A} (DivIVA^{T19A}) and pRGD4A^{T19E} (DivIVA^{T19E}) were grown. The samples were run on 12% SDS-PAGE and immunoblotted with anti-phospho Ser/Thr antibody. *E. coli* harbouring pRadRqkA vector was used as a control (F). *E. coli* BL21 cells were co-transformed with pRadRqkA and pETD4A. Negative controls were only BL21 cells and cells harbouring either pRadRqkA or pETD4A. Expression of RqkA was confirmed by immunoblot using anti-RqkA antibody (G).

Lane 248: At least since the clarification of the DivIVA crystal structure, the "polar determining motif" is and outdated concept. The mentioned residues are for membrane binding. Important literature describing this is not cited (see PMIDs: 30887576, 20502438). Moreover, according to the alignment shown in Fig. S1C, T19 is not part of the "polar determining motif".

Authors' response: Sorry for missing out these references and they are added in the revised manuscript. We have changed polar determining motif with membrane binding motif in the text.

Yes, as you pointed out, T19 does not lie in the polar determining motif and this was mentioned in original submission as well. Text has been modified accordingly.

Fig. S1E: positive control is missing

Authors' response: Thank you. Since RqkA is auto-phosphorylated so the lane pRadRqkA represents the phospho band of RqkA which is considered as positive control in this case. Hope you agree.

Lane 279 onwards: T19E is less expressed than wt DivIVA - I would suspect that this mutation (introduction of a negative charge) alters the structure of the N-terminal lipid binding domain. As a simple test that at least membrane binding and the formation of higher order aggregates are not compromised by T19E, the authors should test oligomerization of wt and T19E DivIVA, for example by blue native PAGE, and membrane binding by cell fractionation (membrane vs cytosolic extracts) with subsequent Western blotting to determine the distribution of DivIVA between the membrane and the cytosol.

Authors' response: Thank you so much for your suggestion. This would be another way to prove both the points raised by reviewer but these were already evidenced under microscopy. For instance, we had provided data on in vivo 3D image of DivIVA^{WT}, DivIVA^{T19A} and DivIVA^{T19E} which would support that the oligomerization of T19E was not affected at least in vivo. For membrane localization, the cells expressing DivIVA^{WT}, DivIVA^{T19A}, and DivIVA^{T19E} were stained with vancomycin and analysed for their co-localization and have found that both these proteins co-localize with the membrane as shown by white dots in the image under the co-localization panel. The respective panels of both the figures are given below for review purpose.

To support oligomerization

Legend: *D. radiodurans* expressing wild type DivIVA-RFP (DivIVA^{WT}) and its phospho-mutants like DivIVA^{T19A} and DivIVA^{T19E} (red), FtsZ-GFP (green) and genome (blue) were analyzed for their planar position using confocal microscopy. The images were taken at several Z-planes and superimposed to give the overall 3-D picture and a single cell was analyzed by rotating at several angles and x-y-z axes are shown in the left bottom corner where they

denote the position of DivIVA, FtsZ and genome respectively. The x-y-z axis is color coded according to the color of DivIVA (**red**), FtsZ (**green**) and nucleoid (**blue**). Data shown is a representative of a reproducible experiment repeated at least 3 times.

To support membrane localization

Legend: Confocal imaging of *D. radiodurans* cells expressing DivIVA^{WT}, DivIVA^{T19A} and DivIVA^{T19E} under constitutive promoter on pRadgro plasmid. The cells (DIC) were stained with fluorescent labelled vancomycin (Van-FL) observed for co-localization of the proteins (red) with the membrane (green) and DivIVA*-RFP where star indicates DivIVA variants (DivIVA^{WT}, DivIVA^{T19A} and DivIVA^{T19E}) and merged panel shows the image combining the two channels. These cells were also analysed for co-localization using the built-in tool in an automated cellSens software, white dot highlights the co-localized pixels of all the proteins with the membrane

Lane 295: The claim that "drDivIVA protein localizes in two planes, (i) in the plane of the FtsZ ring and (ii) in the direction of genome segregation" is an overinterpretation and is not supported by the images shown. Moreover, the statement that "this suggests the possible role of this protein in both cell division and genome segregation by an unknown mechanism" is mere speculation. How can a spotty drDivIVA localization pattern suggest this? To me, the localization pattern seems rather uninformative.

Authors' response: Thank you. Exactly, we were also not able to draw any pattern of DivIVA when it was analysed alone in either 2-D or 3-D planes. That's why we analysed the localization and dynamics pattern of DivIVA and FtsZ protein co-expressing in cell under 3D-view, where we observed that the DivIVA localizes with FtsZ and also in FtsZ free zones (Fig 3) and in 'non-planar view' panel of Fig 5. In response to your comments, the conclusion has been moderated more closer to results.

Lane 302-304: As a role of drDivIVA, its localization and phosphorylation in chromosome segregation and perpendicularity of cell division has not conclusively been shown, this sentence is mere overinterpretation.

Authors' response: Thank you for comments. We have moderated the conclusion.

Lane 309: The chosen citation (43) refers to a review and not to original work. Please cite the work that shows that drdivIVA can only be deleted in the presence of a second copy.

Authors' response: Thank you. The reference [42] has been cited in the manuscript.

Lane 318-321: I find the way to demonstrate essentiality of divIVAT19E not convincing. The authors should construct a mutant with divIVAT19E under control of a constitutive promoter in addition to a wt copy of divIVA under control of an inducible promoter and test this strain for inducer dependency.

Authors' response: Thank you. The suggested experiment has been carried out in response to reviewers comments and shown below for review purpose.

Legend: The wild-type and conditional mutant derivatives were monitored for their growth kinetics using Synergy H1 multi-mode plate reader. The wild-type strain (WT), WT harbouring pVHD4A (WT + D4A), WT harbouring pVHD4A induced with 10 mM IPTG for overnight (WT + D4A (I)), the conditional null mutant of *divIVA* ($\Delta divIVA$) harbouring pVHD4A ($\Delta divIVA$ + D4A), $\Delta divIVA$ + D4A harboring pRGD4A^{T19E} (D4A^{T19E}) were monitored for their growth in the presence and absence of IPTG where I indicates the induced status of the pVHD4A in the corresponding strain as shown in the graph. The growth rate was calculated from the corresponding growth-curve values and plotted using GraphPad Prism software.

Fig. 4D: DivIVA signals in the first two and last three lanes have different molecular weights. Why?

Authors' response: Thank you and sorry for less clarity in legends. The first two lanes of the blot highlight the protein bands from the lysates of the strains where wild type copy of *divIVA* was replaced with *divIVA-rfp* and *divIVA*^{T19A}-*rfp* and the increase in size was due to the addition of *rfp* tag to the protein while other lanes were from the wild type strains where in-trans copy of only *DivIVA* (pRGhisD4A) is present which does not cause the major shift in its size. Hope it is clearer.

Lane 333: The claim that Fig. S2 shows time-lapse data is not true, as different cells at different time points are shown (as stated in the figure legend).

Authors' response: Thank you. It has been corrected in the revision.

Lane 341-343: blue and black arrows are invisible in Fig. 5.

Authors' response: Thank you. The image has been modified.

Lane 343-345: Highly speculative and not supported by data.

Authors' response: Sorry, the conclusion has been modified.

Lane 367-368: I do not see the claim that "the pattern of DivIVA localization has changed..." supported by the microscopic images. I am also not convinced that the quantified data shown in Fig. 6B for wt after irradiation (clear effect) correctly represent the corresponding raw data shown in Fig. 6A (no clear effect). I also have the impression from the data in Fig. 6A-B that DivIVAT19A localization is different from wtDivIVA (more foci), which is in contrast to their earlier claim that DivIVAT19A localizes as wtDivIVA (lanes 347-352).

Authors' response: Thank you for asking us to do further assessment of our results and moderate conclusions. Here, difference or similarity was meant in terms of number of foci and not position of localization. The data shown in Fig 6A is a representative while Fig 6B was generated from a population. We have revised the conclusion. It needs more experiments and better analysis for deeper understanding of foci arrangements in wild type and mutants and will be done independently.

Lane 388-389: "As the structure of the nucleoid changes...genome movement" - Mere overinterpretation, neither "DivIVA movement" nor "DivIVA movement along the direction of chromosome movement" is shown.

Authors' response: The sentences are modified to fit the comment of the reviewer. However, a number of studies with similar data have interpreted results on the line we had done.

Lane 392-393: "Although the pattern of T19E localization ...was evidenced". Again, overinterpretation of data.

Authors' response: The conclusion has been moderated.

January 6, 2023

Dr. Hari S Misra
Bhabha Atomic Research Centre
Molecular Biology Division
GBPant University of Agri and Technology Pantnagar
Anushaktinagar
Pantnagar, UTTAR PRADESH 263145
India

Re: Spectrum03141-22R1 (DivIVA phosphorylation affects its dynamics and cell cycle in radioresistant *Deinococcus radiodurans*)

Dear Dr. Hari S Misra:

Your manuscript has been accepted, and I am forwarding it to the ASM Journals Department for publication. You will be notified when your proofs are ready to be viewed.

Sincerely,

Varsha Singh
Editor, Microbiology Spectrum
